# Nonequilibrium thermodynamics of the asymmetric Sherrington-Kirkpatrick model

Miguel Aguilera [1,2,3] ✉, Masanao Igarashi[4] & Hideaki Shimazaki[5,6]

Most natural systems operate far from equilibrium, displaying time-asymmetric, irreversible dynamics characterized by a positive entropy production while exchanging energy and matter with the environment. Although stochastic thermodynamics underpins the irreversible dynamics of small systems, the nonequilibrium thermodynamics of larger, more complex systems remains unexplored. Here, we investigate the asymmetric Sherrington-Kirkpatrick model with synchronous and asynchronous updates as a prototypical example of large-scale nonequilibrium processes. Using a path integral method, we calculate a generating functional over trajectories, obtaining exact solutions of the order parameters, path entropy, and steady-state entropy production of infinitely large networks. Entropy production peaks at critical order-disorder phase transitions, but is significantly larger for quasi-deterministic disordered dynamics. Consequently, entropy production can increase under distinct scenarios, requiring multiple thermodynamic quantities to describe the system accurately. These results contribute to developing an exact analytical theory of the nonequilibrium thermodynamics of large-scale physical and biological systems and their phase transitions.

While isolated systems tend toward thermodynamic equilibrium, many physical, chemical, and biological processes operate far from equilibrium. Such nonequilibrium systems – from molecules to organisms and machines – persist by exchanging matter and energy with their surroundings, being causally driven by time-varying external stimuli or by their past states (e.g., the adaptive action of sensor and effector interfaces[1]). Nonequilibrium processes inherently break time reversal symmetry, describing spatial and temporal patterns with a definite past-future order, and being thus strikingly different from the reversible dynamics found at thermodynamic equilibrium. Understanding these dissipative processes – from chemical reactions to neural dynamics or flocks of birds – brings critical insights into the self-organization of open systems[2]. Although these ideas have attracted the interest of disparate fields, from evolutionary dynamics[3] to neuroscience[4–7], little is known about the thermodynamic description of nonequilibrium systems comprising many interacting particles.

While stochastic thermodynamics has been greatly influential in the study of small systems with appreciable fluctuations[8], the thermodynamics of large-scale nonequilibrium systems and their phase transitions has attracted attention only very recently[9–11].

When the elements of a system are numerous, characterizing its nonequilibrium states is challenging due to the expansion of its state space. Inspired by the success of the equilibrium Ising model in investigating disordered systems in the thermodynamic limit, we study the nonequilibrium thermodynamics of a stochastic, kinetic Ising model with both synchronous and asynchronous updates. The Ising model is a cornerstone of statistical mechanics, originally conceived as a model describing phase transitions in magnetic materials[12]. A natural extension of the model introducing Markovian dynamics either in discrete or continuous time is the kinetic Ising model, a prototypical model of both equilibrium and nonequilibrium systems such as recurrent neural networks[13] or genetic regulatory networks[14]. With time-independent

[1]BCAM – Basque Center for Applied Mathematics, Bilbao, Spain. [2]IKERBASQUE, Basque Foundation for Science, Bilbao, Spain. [3]School of Engineering and Informatics, University of Sussex, Falmer, Brighton, United Kingdom. [4]Graduate School of Engineering, Hokkaido University, Sapporo, Japan. [5]Graduate School of Informatics, Kyoto University, Kyoto, Japan. [6]Center for Human Nature, Artificial Intelligence, and Neuroscience (CHAIN), Hokkaido University, Sapporo, Japan. ✉e-mail: sci@maguilera.net

parameters and symmetric couplings (under synchronous or asynchronous updates in the absence of lagged self-couplings), the kinetic Ising model results in an equilibrium process exhibiting a variety of complex phenomena, including ordered (ferromagnetic), disordered (paramagnetic), and quenched disordered states (known as spin glasses). The celebrated Sherrington-Kirkpatrick (SK) model, characterized by quenched random couplings resulting in a spin-glass phase[15], can be solved using the replica mean-field method[16]. A kinetic version of this symmetric-coupling model has been represented as a bipartite network, also solved using the replica trick[17].

The kinetics of equilibrium Ising systems are indistinguishable when observed in a forward or backward direction in time, i.e., they are invariant under the reversal of the arrow of time. This time-symmetry breaks down under time-varying external fields or asymmetric couplings comprising history-dependent, non-conservative forces. Such time-asymmetric processes violate detailed balance, leading to nonequilibrium dynamics yielding a positive entropy production[8,18–20]. In the latter case of asymmetric couplings with constant fields, the system may relax towards a steady state known as a nonequilibrium steady state after some time. Time-asymmetric trajectories in steady state are linked with entropy change of the heat baths under 'local detailed balance' for a system coupled to equilibrium reservoirs or heat baths[21,22], suggesting that steady-state entropy production is critical for unveiling the interaction of out-of-equilibrium systems with their environments. Yet, unlike its equilibrium counterpart, the properties of irreversible Ising dynamics remain unclear due to the lack of theoretical description of its entropy production.

Here, we study the kinetics of the SK model with asymmetric connections under synchronous and asynchronous updates as a prototypical model of nonlinear and nonequilibrium processes. As the model does not have a free energy defined in classical terms, we resort to a dynamical equivalent in the form of a generating functional. We apply a path integral approach to obtain exact solutions on its statistical moments and nonequilibrium thermodynamic properties. Unlike the replica method, the generating functional for fully asymmetric couplings has exact solutions in the thermodynamic limit without additional assumptions like analytic continuation and replica symmetry breaking[23]. Previous studies using this method[24,25] have shown that the asymmetric kinetic Ising model with asynchronous updates does not have a spin glass phase. In this manuscript, we will extend the generating functional path integral method to confirm this result in both cases of synchronous and asynchronous updates and further retrieve an exact solution of the entropy production of the system.

One of the open questions in empirical studies is whether an increase in entropy production observed in specific nonequilibrium systems under investigation is linked with the critical properties of systems approaching continuous phase transitions[5,6]. Entropy production is not necessarily maximized under such conditions and can display a continuous change[26], or discontinuities in its derivative[27]. However, a number of simple nonequilibrium systems maximize their entropy production at a critical point. Examples are the entropy production of an Ising model with an oscillatory field and a mean-field majority vote model[28–30]. It is therefore important to investigate the case of the kinetic Ising system as a general model of physical and biological networks. We previously showed that the entropy production of the stationary asymmetric SK model with finite size takes a maximum around a critical point by applying mean-field approximations preserving fluctuations in the system[31]. However, this result (and the aforementioned references) relies on approximations and numerical simulations. Therefore, the assumption that entropy production is maximized near continuous phase transitions has not yet been ratified by exact solutions of spin models in the thermodynamic limit. In this study, we show analytically that the entropy production is locally maximized at critical phase

transition points, representing a potentially useful phase transition correlate for systems without a globally defined free energy or heat capacity. Nevertheless, we also show that entropy production can take larger values for largely heterogeneous couplings in low-temperature regimes exhibiting disordered but nearly deterministic dynamics. Thus, entropy production must be examined carefully, as its increase does not necessarily indicate that the system is approaching a critical state. Instead, combining the entropy rate and entropy production yields a more precise picture of the irreversible processes.

The paper is organized as follows. First, we introduce maximum entropy Markov processes, their entropy production, and a generating functional method used to compute the system's moments and the entropy production in both discrete and continuous time. Next, we describe the asymmetric SK model with synchronous and asynchronous updates and a path integral method calculating the configurational average of the generating functional. This yields an exact solution of the entropy production, magnetization, and correlations in an infinite system. We employ our theoretical results to draw phase diagrams of the order parameters and entropy production for synchronous and asynchronous dynamics with and without randomly sampled external fields. The theoretical predictions are then corroborated by numerical simulation. We also examine the critical line of nonequilibrium phase transitions, the temporal structure of the dynamics, and their relations to the entropy production. Finally, we conclude the paper by discussing the implications of our results for the study of biological systems.

## Results

### Maximum entropy Markov chains

The principle of maximum entropy is a foundation of equilibrium statistical mechanics[32]. The principle has been later generalized for treating time-dependent phenomena, as the principle of maximum caliber or maximum path entropy[33,34]. Under consistency requirements preserving causal interactions, the maximum caliber principle yields a Markov process[35]. To see this, we start with a discrete-time stochastic process with $N$ discrete-state elements defined at time $u$ as $\mathbf{s}_u = \{s_{1,u}, \ldots, s_{N,u}\}$ for discrete-time trajectories of length $t + 1$ defined by a path probability $p(\mathbf{s}_{0:t})$. Later, we will show this discrete-time formulation can be generalized to an equivalent continuous-time formulation under appropriate assumptions.

Path entropy is defined as

$$S_{0:t} = -\sum_{\mathbf{s}_{0:t}} p(\mathbf{s}_{0:t}) \log p(\mathbf{s}_{0:t}). \tag{1}$$

Maximizing Eq. (1), subject to constraints, yields the least structured distribution $p(\mathbf{s}_{0:t})$ consistent with observations[36]. In causal network models, entropy maximization has to be constrained with a set of temporal consistency requirements[35], as was first established by Kolmogorov[37]. Specifically, for any positive integer $u(\leq t)$, we impose

$$\sum_{\mathbf{s}_u} p_{0:u}(\mathbf{s}_{0:u}) = p_{0:u-1}(\mathbf{s}_{0:u-1}), \tag{2}$$

where $p_{0:u}(\mathbf{s}_{0:u})$ is given by

$$p_{0:u}(\mathbf{s}_{0:u}) = p(\mathbf{s}_{0:u}) \arg\max S_{0:u}. \tag{3}$$

That is, we impose consistency between the marginal distribution for the maximum entropy path $\mathbf{s}_{0:u-1}$ in $p_{0:u}(\mathbf{s}_{0:u})$ and the maximum entropy distribution of path $\mathbf{s}_{0:u-1}$, $p_{0:u-1}(\mathbf{s}_{0:u-1})$. This constrains path distribution dependencies between consecutive states. We will drop the subscript in the path probability when not needed.

Maximizing Eq. (1) with constraints $f_n(\mathbf{s}_u, \mathbf{s}_{u-1}) = C_{n,u}$ (where $C_{n,u}$ is a constant for the $n$-th constraint at time $u$), an initial distribution $p(\mathbf{s}_0)$, and Eq. (2) results in a Markovian process[35]

$$p(\mathbf{s}_{0:t}) = p(\mathbf{s}_0) \prod_{u=1}^{t} p(\mathbf{s}_u|\mathbf{s}_{u-1})$$
$$\propto p(\mathbf{s}_0) \prod_{u=1}^{t} \exp\left[\sum_n \lambda_n f_n(\mathbf{s}_u, \mathbf{s}_{u-1})\right]. \tag{4}$$

The path entropy can be then decomposed into

$$S_{0:t} = -\sum_{\mathbf{s}_{0:t}} p(\mathbf{s}_{0:t})\left(\sum_u \log p(\mathbf{s}_u|\mathbf{s}_{u-1}) + \log p(\mathbf{s}_0)\right)$$
$$= \sum_u S_{u|u-1} + S_0, \tag{5}$$

where $S_0$ is the entropy of the initial distribution and $S_{u|u-1}$ is a conditional entropy, defined as

$$S_{u|u-1} = -\sum_{\mathbf{s}_u, \mathbf{s}_{u-1}} p(\mathbf{s}_u, \mathbf{s}_{u-1}) \log p(\mathbf{s}_u|\mathbf{s}_{u-1}), \tag{6}$$

which, at the steady state described in the following, corresponds to the Kolmogorov-Sinai entropy or entropy rate, $\lim_{t\to\infty} \frac{1}{t} S_{0:t}$.

## Nonequilibrium steady state

A Markov chain converges to a unique stationary distribution if the system is irreducible (all states are accessible from any state in finite time) and aperiodic (the greatest common divisor of the number of steps for returning to the same state with non-zero probability is one[38]). We can confirm that these requirements are satisfied by Eq. (4) with finite transition probabilities, thus warranting the existence of a steady-state distribution $\pi(\mathbf{s}_u)$, which can be either in or out of thermodynamic equilibrium, as explained in the following.

For a discrete-time Markov chain, the evolution of the state probability distribution follows a master equation:

$$p_u(\mathbf{s}_u) = \sum_{\mathbf{s}_{u-1}} p(\mathbf{s}_u|\mathbf{s}_{u-1}) p_{u-1}(\mathbf{s}_{u-1})$$
$$= p_{u-1}(\mathbf{s}_u) + \sum_{\mathbf{s}_{u-1}} j^u_{\mathbf{s}_{u-1}\to\mathbf{s}_u}. \tag{7}$$

Here $p_v(\mathbf{s}_u)$ is a marginal probability distribution of a state $\mathbf{s}_u$ calculated for the distribution at time $v$. For simplicity, we will omit the subscript and write $p(\mathbf{s}_u)$ when $v = u$. $j^u_{\mathbf{s}_{u-1}\to\mathbf{s}_u}$ are the system's probability fluxes:

$$j^u_{\mathbf{s}_{u-1}\to\mathbf{s}_u} \equiv p(\mathbf{s}_u|\mathbf{s}_{u-1})p(\mathbf{s}_{u-1}) - p(\mathbf{s}_{u-1}|\mathbf{s}_u)p_{u-1}(\mathbf{s}_u). \tag{8}$$

In the limit of small probability fluxes, the system can be described by an equivalent continuous-time process:

$$\frac{dp(\mathbf{s},t)}{dt} = \sum_{\mathbf{s}'} j_{\mathbf{s}'\to\mathbf{s}}(t) \tag{9}$$

$$j_{\mathbf{s}'\to\mathbf{s}}(t) \equiv w(\mathbf{s}|\mathbf{s}')p(\mathbf{s}',t) - w(\mathbf{s}'|\mathbf{s})p(\mathbf{s},t), \tag{10}$$

where $t$ refers to the continuous time and $w(\mathbf{s}|\mathbf{s}')$ are transition rates.

The system is stationary or in a steady state if the sum of all probability fluxes is zero for all $\mathbf{s}_u$, i.e., $\sum_{\mathbf{s}_u,\mathbf{s}_{u-1}} j^u_{\mathbf{s}_{u-1}\to\mathbf{s}_u} = 0$. In addition, this will be an equilibrium steady state if $j^u_{\mathbf{s}_{u-1}\to\mathbf{s}_u} = 0$ for all pairs $\mathbf{s}_{u-1}$, $\mathbf{s}_u$,

resulting in the detailed balance condition

$$p(\mathbf{s}_u|\mathbf{s}_{u-1})\pi(\mathbf{s}_{u-1}) = p(\mathbf{s}_{u-1}|\mathbf{s}_u)\pi(\mathbf{s}_u), \tag{11}$$

where $\pi(\mathbf{s}_u)$ is the steady-state distribution. When detailed balance is broken under the stationary condition, i.e., some $j^u_{\mathbf{s}_{u-1}\to\mathbf{s}_u} \neq 0$ but their sum is equal to zero, the stationary system is in a nonequilibrium steady state.

## Steady-state entropy production

Stochastic thermodynamics describes a link between the time-irreversible stochastic trajectories with surroundings in the form of heat (entropy) dissipation. As the system evolves, it experiences an entropy change $\sigma_u^{\text{sys}}$:

$$\sigma_u^{\text{sys}} = S_u - S_{u-1} = \sum_{\mathbf{s}_u,\mathbf{s}_{u-1}} p(\mathbf{s}_u,\mathbf{s}_{u-1}) \log\frac{p(\mathbf{s}_{u-1})}{p(\mathbf{s}_u)}. \tag{12}$$

Nonequilibrium systems maintain irreversible dynamics by continuously dissipating heat (entropy) to their environments. Under local detailed balance[21,22,39] in a system coupled to a heat bath, the entropy change results from subtracting the entropy dissipated to the heat bath $\sigma_u^{\text{bath}}$ from the (total) entropy production $\sigma_u$:

$$\sigma_u^{\text{sys}} = \sigma_u - \sigma_u^{\text{bath}}, \tag{13}$$

where the entropy change of the heat bath is given as

$$\sigma_u^{\text{bath}} = \sum_{\mathbf{s}_u,\mathbf{s}_{u-1}} p(\mathbf{s}_u,\mathbf{s}_{u-1}) \log\frac{p(\mathbf{s}_u|\mathbf{s}_{u-1})}{p(\mathbf{s}_{u-1}|\mathbf{s}_u)}. \tag{14}$$

Here $p(\mathbf{s}_{u-1}|\mathbf{s}_u)$ is a transition probability (from Eq. (4)) but evaluated by the reverse trajectory[21,40], that is, we define it using the transition function at time $u$, but switch $\mathbf{s}_u$ and $\mathbf{s}_{u-1}$. This equation relates the system's time asymmetry with the entropy change of the reservoir.

The entropy production $\sigma_u$ at time $u$ is then given as

$$\sigma_u = \sum_{\mathbf{s}_u,\mathbf{s}_{u-1}} p(\mathbf{s}_u,\mathbf{s}_{u-1}) \log\frac{p(\mathbf{s}_u|\mathbf{s}_{u-1})p(\mathbf{s}_{u-1})}{p(\mathbf{s}_{u-1}|\mathbf{s}_u)p(\mathbf{s}_u)}, \tag{15}$$

which is the Kullback–Leibler divergence between the forward and backward trajectories[8,18,20,41]. Due to the non-negativity of the divergence, the entropy production is non-negative, $\sigma_u \geq 0$. This entropy production vanishes if the probability of forward trajectories is identical to a posterior of past states given the future state[20], i.e., when the process loses time-asymmetry in prediction and postdiction[42].

Alternatively, the dissipation function[8,43,44] quantifies the difference between incoming and outgoing fluxes in Eq. (8):

$$\widetilde{\sigma}_u = \sum_{\mathbf{s}_u,\mathbf{s}_{u-1}} p(\mathbf{s}_u,\mathbf{s}_{u-1}) \log\frac{p(\mathbf{s}_u|\mathbf{s}_{u-1})p(\mathbf{s}_{u-1})}{p(\mathbf{s}_{u-1}|\mathbf{s}_u)p_{u-1}(\mathbf{s}_u)}, \tag{16}$$

which directly assesses a violation of the detailed balance. The entropy production $\sigma_u$ and dissipation function $\widetilde{\sigma}_u$ are equivalent under steady-state conditions. Furthermore, both quantities become equivalent in the continuous-time limit[8,42,44] and converge to the entropy production rate[39]:

$$\frac{d\sigma(t)}{dt} = \frac{1}{2}\sum_{\mathbf{s},\mathbf{s}'} j_{\mathbf{s}'\to\mathbf{s}}(t) \log\frac{w(\mathbf{s}|\mathbf{s}')p(\mathbf{s}',t)}{w(\mathbf{s}'|\mathbf{s})p(\mathbf{s},t)}. \tag{17}$$

In a steady state, the entropy production is caused by dissipation only and becomes equivalent to the *house-keeping* entropy production caused by the non-conservative forces under a steady state[45,46]. Both $\sigma_u$

and $\widetilde{\sigma}_u$ result in:

$$\sigma_u = \widetilde{\sigma}_u = \sigma_u^{\text{bath}} = -S_{u|u-1} + S_{u|u-1}^r. \tag{18}$$

Here $S_{u|u-1}^r$ is the entropy of the time-reversed conditional distribution:

$$S_{u|u-1}^r \equiv -\sum_{\mathbf{s}_u,\mathbf{s}_{u-1}} p(\mathbf{s}_u,\mathbf{s}_{u-1}) \log p(\mathbf{s}_{u-1}|\mathbf{s}_u). \tag{19}$$

In this paper, we study the steady-state entropy production in Eq. (18), which is critical for evaluating the interaction of the nonequilibrium processes with their environment.

### Generating functional

Consider a maximum caliber path probability (Eq. (4))

$$p(\mathbf{s}_{0:t}) = \prod_{u=1}^{t} p(\mathbf{s}_u|\mathbf{s}_{u-1})p(\mathbf{s}_0), \tag{20}$$

For simplicity, we will assume $p_0(\mathbf{s}) = \prod_i \delta[s_i, s_{i,0}]$ – the initial distribution is a Kronecker delta with a unique initial state – and ignore the term. However, the following steps are general for any $p_0(\mathbf{s}_0)$.

In equilibrium systems, the partition function retrieves their statistical moments. A nonequilibrium equivalent function is a generating functional or dynamical partition function. To obtain not only the statistical properties averaged over trajectories, but also the forward/time-reversed conditional entropies (Eqs. (6), (19)), we define the following generating functional:

$$Z_t(\mathbf{g}) = \sum_{\mathbf{s}_{0:t}} p(\mathbf{s}_{0:t}) \exp\left[\Gamma(\mathbf{g}, \mathbf{s}_{0:t})\right], \tag{21}$$

$$\Gamma(\mathbf{g}, \mathbf{s}_{0:t}) = \sum_{i,u} g_{i,u} s_{i,u} - \sum_u g_u^S \epsilon(\mathbf{s}_u|\mathbf{s}_{u-1}) \\ - \sum_u g_u^{S^r} \epsilon(\mathbf{s}_{u-1}|\mathbf{s}_u), \tag{22}$$

where $\epsilon(\mathbf{s}_u|\mathbf{s}_{u-1}) \equiv -\log p(\mathbf{s}_u|\mathbf{s}_{u-1})$. In the limit $t \to \infty$, the logarithm of the generating functional converges to the large deviation function[47–49],

$$\varphi(\mathbf{g}) = \lim_{t \to \infty} \frac{1}{t} \log Z_t(\mathbf{g}), \tag{23}$$

which plays the role of a free-energy function for nonequilibrium trajectories[50]. The vector $\mathbf{g}$ is composed of parameters $g_{i,u}$ ($i = 1, ..., N, u = 1, ..., t$) and $g_u^S, g_u^{S^r}$ ($u = 1, ..., t$) retrieving the system's statistical properties. The parameters $g_{i,u}$ recover the statistical moments of the systems like the average rates and correlations:

$$\lim_{\mathbf{g} \to \mathbf{0}} \frac{\partial Z_t(\mathbf{g})}{\partial g_{i,u}} = \lim_{\mathbf{g} \to \mathbf{0}} \langle s_{i,u} \rangle_{\mathbf{g}} = \langle s_{i,u} \rangle, \tag{24}$$

$$\lim_{\mathbf{g} \to \mathbf{0}} \frac{\partial^2 Z_t(\mathbf{g})}{\partial g_{i,u} \partial g_{j,v}} = \lim_{\mathbf{g} \to \mathbf{0}} \left\langle s_{i,u} s_{j,v} \right\rangle_{\mathbf{g}} = \left\langle s_{i,u} s_{j,v} \right\rangle, \tag{25}$$

where angle brackets are defined as

$$\langle f(\mathbf{s}_{0:t}) \rangle_{\mathbf{g}} = \sum_{\mathbf{s}_{0:t}} f(\mathbf{s}_{0:t}) \exp\left[\sum_{i,u} g_{i,u} s_{i,u}\right] p(\mathbf{s}_{0:t}), \tag{26}$$

$$\langle f(\mathbf{s}_{0:t}) \rangle = \sum_{\mathbf{s}_{0:t}} f(\mathbf{s}_{0:t}) p(\mathbf{s}_{0:t}). \tag{27}$$

In addition, $g_u^S, g_u^{S^r}$ retrieve the conditional and time-reversed conditional entropy terms, $S_{u|u-1}, S_{u|u-1}^r$:

$$S_{u|u-1} = -\lim_{\mathbf{g} \to \mathbf{0}} \frac{\partial Z_t(\mathbf{g})}{\partial g_u^S} \\ = \lim_{\mathbf{g} \to \mathbf{0}} \langle \epsilon(\mathbf{s}_u|\mathbf{s}_{u-1}) \rangle_{\mathbf{g}} = \langle \epsilon(\mathbf{s}_u|\mathbf{s}_{u-1}) \rangle, \tag{28}$$

$$S_{u|u-1}^r = -\lim_{\mathbf{g} \to \mathbf{0}} \frac{\partial Z_t(\mathbf{g})}{\partial g_u^{S^r}} \\ = \lim_{\mathbf{g} \to \mathbf{0}} \langle \epsilon(\mathbf{s}_{u-1}|\mathbf{s}_u) \rangle_{\mathbf{g}} = \langle \epsilon(\mathbf{s}_{u-1}|\mathbf{s}_u) \rangle, \tag{29}$$

and thus the steady-state entropy production (Eq. (18)):

$$\sigma_u = \lim_{\mathbf{g} \to \mathbf{0}} \left( \frac{\partial Z_t(\mathbf{g})}{\partial g_u^S} - \frac{\partial Z_t(\mathbf{g})}{\partial g_u^{S^r}} \right). \tag{30}$$

### Synchronous and asynchronous, asymmetric Sherrington-Kirkpatrick model

We consider $N$ interacting elements $\mathbf{s}_u$ (spins or neurons), taking each element $i$ at time $u$ a binary state $s_{i,u} = \{-1, 1\}$. Constraints take the form of delayed pairwise couplings (i.e., $f_{ij}(\mathbf{s}_u, \mathbf{s}_{u-1}) = s_{i,u} s_{j,u-1}$ in Eq. (4)). This results in the dynamics:

$$p(\mathbf{s}_u|\mathbf{s}_{u-1}) = \prod_i \frac{\exp[\beta s_{i,u} h_{i,u}]}{2 \cosh[\beta h_{i,u}]}, \tag{31}$$

$$h_{i,u} = H_{i,u} + \sum_j J_{ij} s_{j,u-1}, \tag{32}$$

where $\beta$ is the inverse temperature. The system's state at time $u$ depends on the previous time-step (Fig. 1a).

The equation above is a general formulation of a kinetic Ising model with time-dependent fields $H_{i,u}$. The dynamics can include both synchronous and asynchronous Ising systems by introducing a set of independent Bernoulli random variables: $\tau_{i,u} = 0, 1$ with probabilities $1 - \alpha$ and $\alpha$ (i.e., $\tau_{i,u} \sim \text{Bernoulli}(\alpha)$) and making $H_{i,u}$ stochastic processes:

$$H_{i,u} = \Theta_{i,u} + (1 - \tau_{i,u})K s_{i,u-1}. \tag{33}$$

Note that in the limit of $K \to \infty$, the state $s_{i,u}$ is tightly coupled to the previous state $s_{i,u-1}$. Therefore, the state changes only if $\tau_{i,u} = 1$. We have the following transition probability in the $K \to \infty$ limit:

$$p(\mathbf{s}_u|\mathbf{s}_{u-1}) = \prod_i (\tau_{i,u} w(s_{i,u}|\mathbf{s}_{u-1}) \\ + (1 - \tau_{i,u}) \delta[s_{i,u}, s_{i,u-1}]), \tag{34}$$

with the transition rate

$$w(s_{i,u}|\mathbf{s}_{u-1}) = \frac{\exp[\beta s_{i,u} h_{i,u}^1]}{2 \cosh[\beta h_{i,u}^1]}, \tag{35}$$

where $h_{i,u}^1 = \Theta_{i,u} + \sum_j J_{ij} s_{j,u-1}$ and $\Theta_{i,u}$ is an external field. With $\alpha = 1$ we have a kinetic Ising system under parallel or synchronous updates. In

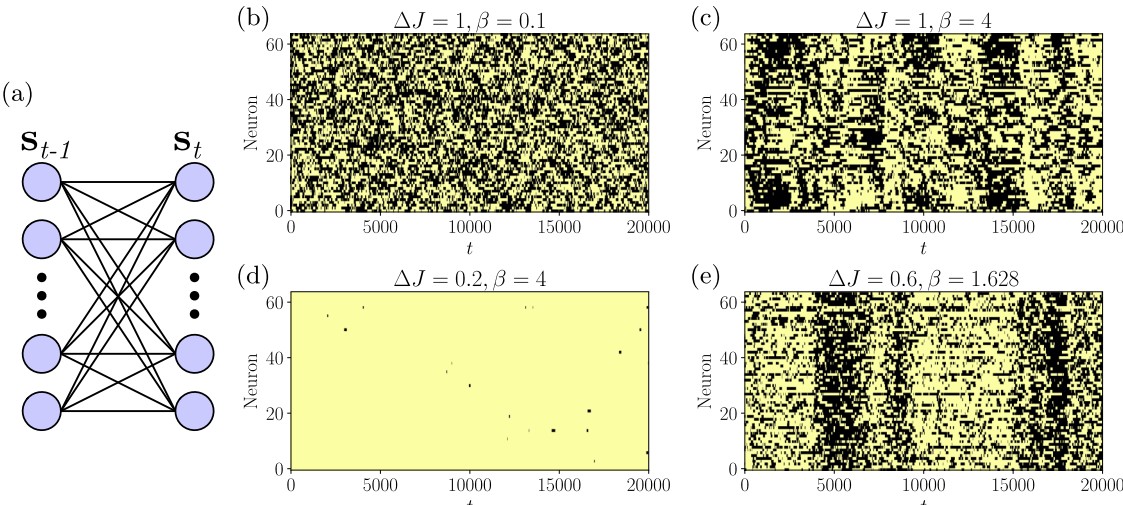

**Fig. 1 | Asymmetric kinetic SK model. a** The asymmetric kinetic Ising model describes a Markov chain where states at time $\mathbf{s}_u$ depend on pairwise couplings to states $\mathbf{s}_{u-1}$. This model shows disordered dynamics for large coupling variance both at high and low temperatures (**b** and **c**), ordered dynamics for low temperatures and a low coupling variance (**d**), and critical dynamics at the phase transition (**e**).

the limit $\alpha \to 0$, we have in turn a kinetic Ising system with asynchronous updates (i.e., at most one spin is updated each time step), converging to a continuous-time master equation.

The generating functional of the kinetic Ising system (Eq. (21)) is defined by the functions

$$\epsilon(\mathbf{s}_u | \mathbf{s}_{u-1}) = -\sum_i \left(\beta s_{i,u} h_{i,u} + \log[2\cosh[\beta h_{i,u}]]\right), \quad (36)$$

$$\epsilon(\mathbf{s}_{u-1} | \mathbf{s}_u) = -\sum_i \left(\beta s_{i,u-1} h_{i,u}^r + \log[2\cosh[\beta h_{i,u}^r]]\right), \quad (37)$$

where $h_{i,u}^r = H_{i,u} + \sum_j J_{ij} s_{j,u} = h_{i,u+1} + H_{i,u} - H_{i,u+1}$.

The equilibrium Ising model with symmetric random Gaussian couplings is referred to as the SK model. In the fully-asymmetric SK model, the couplings $J_{ij}$ are quenched independent variables, each following a Gaussian distribution

$$p(J_{ij}) = \frac{1}{\sqrt{2\pi\Delta J^2/N}} \exp\left[\frac{-1}{2\Delta J^2/N}\left(J_{ij} - \frac{J_0}{N}\right)^2\right], \quad (38)$$

with mean $J_0/N$ and variance $\Delta J^2/N$ scaled by $N$.

The asymmetric SK model shows a variety of population dynamics. Figure 1 shows exemplary dynamics under asynchronous updates without fields ($\Theta_{i,u} = 0$). It shows disordered dynamics for large coupling variance $\Delta J^2$ both at high and low temperatures (i.e. low and large $\beta$, Fig. 1b, c), ordered dynamics for low temperatures and low $\Delta J^2$ (Fig. 1d), and critical dynamics at the phase transition (Fig. 1e).

## Solution of the asymmetric Sherrington-Kirkpatrick model

The solution of the kinetic version of the SK model with asymmetric and quenched couplings can be obtained by computing the generating functional averaged over the couplings (referred to as the configurational average):

$$[Z_t(\mathbf{g})]_{\mathbf{J}} = \int d\mathbf{J} Z_t(\mathbf{g}) \prod_{ij} p(J_{ij}). \quad (39)$$

This integral cannot be solved directly because of the $\log[2\cosh[\cdot]]$ terms in Eqs. (36) and (37), which depend nonlinearly on $J_{ij}$. A path

integral method[51] to find a solution introduces a delta integral representing $\beta h_{i,u}$ with auxiliary variables $\theta_{i,u} = \beta(H_{i,u} + \sum_j J_{ij} s_{j,u-1})$ as well as $\beta h_{i,u}^r$ with an auxiliary variable $\vartheta_{i,u} = \theta_{i,u+1} + \beta(H_{i,u} - H_{i,u+1})$. Let $\boldsymbol{\theta} = \{\theta_{i,u}\}$ (note $u = 1, \ldots, t+1$) and $\boldsymbol{\vartheta} = \{\vartheta_{i,u}\}$ ($u = 0, \ldots, t$) denote a set of the auxiliary variables. Using conjugate variables $\hat{\boldsymbol{\theta}} = \{\hat{\theta}_{i,u}\}$ to represent the delta function in the integral form, the configurational average is written as

$$[Z_t(\mathbf{g})]_{\mathbf{J}} = \frac{1}{(2\pi)^{N(t+1)}} \int d\boldsymbol{\theta} d\hat{\boldsymbol{\theta}} d\mathbf{J} \left(\prod_{ij} p(J_{ij})\right)$$
$$\cdot \sum_{\mathbf{s}_{1:t}} \exp\left[\sum_{i,u}(s_{i,u}\theta_{i,u} - \log[2\cosh\theta_{i,u}])\right.$$
$$\left. + \sum_{i,u} i\hat{\theta}_{i,u}\left(\theta_{i,u} - \beta H_{i,u} - \beta \sum_j J_{ij} s_{j,u-1}\right) + \Gamma(\mathbf{g}, \mathbf{s}_{0:t}, \boldsymbol{\theta}, \boldsymbol{\vartheta})\right], \quad (40)$$

with

$$\Gamma(\mathbf{g}, \mathbf{s}_{0:t}, \boldsymbol{\theta}, \boldsymbol{\vartheta}) = \sum_{i,u} \Gamma_{i,u}(\mathbf{g}, \mathbf{s}_{0:t}, \theta_{i,u}, \vartheta_{i,u})$$
$$= \sum_{i,u} \left(g_{i,u} s_{i,u} + g_u^s(s_{i,u}\theta_{i,u} - \log[2\cosh[\theta_{i,u}]])\right. \quad (41)$$
$$\left. + g_u^{s^r}(s_{i,u-1}\vartheta_{i,u} - \log[2\cosh[\vartheta_{i,u}]])\right).$$

Note that the summation of $\hat{\theta}_{i,u}$ terms is performed over $u = 1, \ldots, t+1$ to retrieve the fields of both the forward and backward trajectories.

The integral over $J_{ij}$ can be now performed directly over linear exponential terms (see Supplementary Note 1). After integration, Eq. (40) incorporates quadruple-wise interactions among spins $\mathbf{s}_{0:t}$ and conjugate variables $\hat{\boldsymbol{\theta}}$ (Supplementary Eq. (S1.10)), similar to replica interactions in the equilibrium SK model[12]. These interactions are simplified by introducing Gaussian integrals and a saddle-point approximation in the thermodynamic limit (Supplementary Eq. (S1.26)). The saddle-point solution can be simplified by introducing four types of order parameters (Supplementary Eq. (S1.31)). In fully-asymmetric networks, two of these order parameters are found to be zero at $\mathbf{g} = \mathbf{0}$, yielding a solution in terms of the order parameters $m_u$

and $q_{u,v}$ (see Supplementary Eqs. (S1.48) and (S1.49)):

$$m_u = \frac{1}{N}\sum_i \left[ \langle s_{i,u} \rangle \right]_{\mathbf{J}}, \qquad (42)$$

$$q_{u,v} = \frac{1}{N}\sum_i \left[ \langle s_{i,u} s_{i,v} \rangle \right]_{\mathbf{J}}. \qquad (43)$$

Finally, conjugate variables $\hat{\boldsymbol{\theta}}$ in the saddle-point solution can be substituted with a multivariate Gaussian integral (Supplementary Eq. (S1.57)), leading to a factorized generating functional

$$
\begin{aligned}
\left[ Z_t(\mathbf{g}) \right]_{\mathbf{J}} = \prod_i \sum_{\mathbf{s}_{i,1:t}} \int d\boldsymbol{\xi}\, p(\boldsymbol{\xi}) \exp\Big[ & \sum_u s_{i,u} \overline{h}_{i,u}(\xi_u) \\
& + \sum_u \beta s_{i,u-1} \widetilde{h}_{i,u-1} - \sum_u \log 2\cosh\left[ \beta \overline{h}_{i,u}(\xi_u) \right] \\
& + \sum_u \Gamma_{i,u}(\mathbf{g}, \mathbf{s}_{0:t}, \beta \overline{h}_{i,u}(\xi_u), \beta \overline{h}^r_{i,u}(\xi_{u+1})) \Big],
\end{aligned}
\qquad (44)
$$

where the stochastic elements $\boldsymbol{\xi} = (\xi_1, \ldots, \xi_{t+1})$ affecting each spin $i$ follow a multivariate normal distribution $p(\boldsymbol{\xi}) = \mathcal{N}(\mathbf{0}, \mathbf{q})$ with $\mathbf{q}$ defining $q_{u-1,v-1}$ as the covariance of each pair $\xi_u, \xi_v$ for $u, v \in 1, \ldots, t+1$. Here, at $\mathbf{g} = \mathbf{0}$, spin interactions are effectively substituted by same-spin temporal couplings in mean effective fields

$$\overline{h}_{i,u}(\xi_u) = H_{i,u} + J_0 m_{u-1} + \Delta J \xi_u, \qquad (45)$$

$$\overline{h}^r_{i,u}(\xi_{u+1}) = H_{i,u} + J_0 m_u + \Delta J \xi_{u+1}, \qquad (46)$$

$$\widetilde{h}_{i,u-1} = 0. \qquad (47)$$

Applying Eqs. (24) and (25) to the configurational average in Eq. (44), we obtain the order parameters $m_u$ and $q_{u,v}$:

$$m_u = \frac{1}{N}\sum_i \int \mathrm{D}z \tanh\left[ \beta \overline{h}_{i,u}(z) \right], \qquad (48)$$

$$q_{u,v} = \frac{1}{N}\sum_i \int \mathrm{D}xy^{(q_{u-1,v-1})} \tanh\left[ \beta \overline{h}_{i,u}(x) \right] \cdot \tanh\left[ \beta \overline{h}_{i,v}(y) \right], \qquad (49)$$

where the Gaussian stochastic terms are simplified to

$$\mathrm{D}z = \frac{1}{\sqrt{2\pi}} \exp\left[ -\frac{1}{2}z^2 \right], \qquad (50)$$

$$\mathrm{D}xy^{(q_{u,v})} = \frac{1}{2\pi\sqrt{1-q_{u,v}^2}} \exp\left[ \frac{-x^2 - y^2 + 2q_{u,v}xy}{2(1-q_{u,v}^2)} \right]. \qquad (51)$$

In contrast with the equilibrium SK model, $m_u$ is independent of $q_{u,v}$, resulting in the lack of a spin-glass phase as suggested by previous studies[25].

The configurational average of Eqs. (28) and (29) results in the following conditional entropy and time-reversed conditional entropy

$$
\begin{aligned}
\left[ S_{u|u-1} \right]_{\mathbf{J}} = \sum_i \int -\mathrm{D}z \Big( & \beta(H_{i,u} + J_0 m_{u-1}) \tanh\left[ \beta \overline{h}_{i,u}(z) \right] \\
& + \beta^2 \Delta J^2 \left( 1 - \tanh^2\left[ \beta \overline{h}_{i,u}(z) \right] \right) \\
& - \log\left[ 2\cosh\left[ \beta \overline{h}_{i,u}(z) \right] \right] \Big),
\end{aligned}
\qquad (52)
$$

$$
\begin{aligned}
\left[ S^r_{u|u-1} \right]_{\mathbf{J}} = \sum_i \int -\mathrm{D}z \Big( & \beta(H_{i,u} + J_0 m_u) \tanh\left[ \beta \overline{h}_{i,u-1}(z) \right] \\
& + \beta^2 \Delta J^2 q_{u,u-2} \left( 1 - \tanh^2\left[ \beta \overline{h}_{i,u-1}(z) \right] \right) \\
& - \log\left[ 2\cosh\left[ \beta \overline{h}^r_{i,u}(z) \right] \right] \Big).
\end{aligned}
\qquad (53)
$$

Up to this point, our results are general for time-dependent fields $H_{i,u}$, covering synchronous and asynchronous updates by Eq. (33). We obtain the results for the synchronous SK model by setting $\alpha = 1$ or, equivalently, $H_{i,u} = \Theta_{i,u}$. For time-independent fields ($\Theta_{i,u} = \Theta_i$), the system converges to a steady state determined by the solution of the self-consistent equations given by Eqs. (48) and (49). Finally, using Eq. (30), the steady-state entropy production under the synchronous updates is obtained as

$$\left[ \sigma_u \right]_{\mathbf{J}} = \beta^2 \Delta J^2 (1 - q_{u,u-2}) \sum_i \int \mathrm{D}z (1 - \tanh^2\left[ \beta\left( \Theta_i + J_0 m_{u-1} + \Delta J z \right) \right]), \qquad (54)$$

with $m_{u-1}$ and $q_{u,u-2}$ given by their steady-state values (i.e., independent of $u$). Note that for the synchronous system the steady-state solution of $q_{u,v}$ is the same for all $u, v$. In the following, we will use $m$ and $q$ to represent these steady-state solutions.

To calculate the steady-state solutions for the asynchronous SK model, we calculate the generating functional $\left[ Z_t(\mathbf{g}) \right]_{\mathbf{J},\boldsymbol{\tau}}$ that is additionally averaged over the independent random variables $\tau_{i,u}$ in Eq. (33). We show in Supplementary Note 2 that the resulting order parameters in continuous-time $m(t)$ and $q(t',t)$ are subject to the following dynamical equations:

$$\frac{dm(t)}{dt} = \frac{1}{N}\sum_i \int \mathrm{D}z \tanh\left[ \beta h^*_i(z,t) \right] - m(t). \qquad (55)$$

$$\frac{dq(t',t)}{dt} = q^1(t',t) - q(t',t), \qquad (56)$$

$$\frac{dq^1(t',t)}{dt'} = \frac{1}{N}\sum_i \int \mathrm{D}xy^{(q(t',t))} \tanh\left[ \beta h^*_i(x,t') \right] \cdot \tanh\left[ \beta h^*_i(y,t) \right] - q^1(t',t) \qquad (57)$$

with $h^*_i(z,t) = \Theta_i + J_0 m(t) + \Delta J z$. Here $q^1(t',t)$ is the spin correlation conditioned on spins being updated at time $t$. The steady-state solutions of $m(t)$ and $q(t',t)$ (assuming $t' \gg t$) converge to the same steady-state values $m$ and $q$ found for the synchronous SK model (see Supplementary Note 2).

In the continuous-time limit, the steady-state entropy production converges to a steady-state entropy production rate (Eq. (17)) given by:

$$\left[ \frac{d\sigma(t)}{dt} \right]_{\mathbf{J},\boldsymbol{\tau}} = \lim_{\alpha\to 0} \beta^2 \Delta J^2 (1 - q(t+\alpha, t-\alpha)) \cdot \sum_i \int \mathrm{D}z \left( 1 - \tanh^2\left[ \beta h^*_i(z,t) \right] \right). \qquad (58)$$

The delayed-self correlation $q(t',t)$ is discontinuous at $t' = t$ (i.e., $\lim_{\alpha\to 0} q(t+\alpha, t-\alpha) \neq q(t,t) = 1$, see Supplementary Fig. S1), warranting that the entropy production rate can be non-zero for appropriate parameters.

Given the analytical solutions of the system, we will now study the phase diagram of the SK model. In contrast with the naive replica-symmetric solution of the equilibrium SK model, the equations above

are exact in the model with asymmetric couplings in the thermo-dynamic limit under both synchronous and asynchronous updates.

## The SK model without external fields

Figure 2a, b displays the phase diagram of the steady-state order parameters, $m$ and $q$, for both synchronous and asynchronous updates, respectively derived from Eqs. (48) and (49) as a function of the inverse temperature $\beta$ and the width of the coupling distribution $\Delta J$, when the external fields are fixed at zeros ($\Theta_{i,u} = 0$) and the mean coupling is $J_0 = 1$. In this setting, the inverse temperature $\beta$ controls the magnitude of the couplings. The phase diagram shows two distinct regions, one in which the order parameters are fixed at zero (zero magnetization and zero self-correlations, $m = 0$ and $q = 0$) – indicating disordered states – and the other in which the order parameters become positive ($m > 0$ and $q > 0$) – indicating ordered states. Therefore, the system exhibits a nonequilibrium analogue of the paramagnetic-ferromagnetic (disorder-order) phase transition controlled by the parameters, $\beta$ and $\Delta J$. The dashed line in each panel shows the critical values of $\Delta J$ as a function of $\beta$, which is obtained by solving the following equation (see Supplementary Note 3),

$$\frac{1}{\beta J_0} = \int Dz \left(1 - \tanh^2[\beta(\Delta J z)]\right). \tag{59}$$

The solution will be denoted as $\Delta J^c(\beta)$. As studied in Supplementary Note 3, this critical phase transition corresponds to the mean-field universality class, as in the order-disorder phase transition of the equilibrium SK model (note that the spin-glass phase has different exponents[52]).

Depending on the coupling variance $\Delta J$, the dynamics do or do not undergo the nonequilibrium phase transition by varying the inverse temperature $\beta$. The critical $\Delta J^c(\beta)$ at $\beta \to \infty$ is given as $\Delta J^c(\infty) = 0.79501$ (dotted horizontal line). If the distribution is narrower than the critical value $\Delta J^c(\infty)$, the process undergoes the phase transition by changing $\beta$. If the distribution is wider than the critical value, the order parameters are fixed at zeros ($m = 0, q = 0$) for any $\beta$. Note that, for $\beta \to \infty$ (zero temperature), the activation function approaches the threshold non-linearity given by the sign function; therefore, the process becomes deterministic. That is, for the large values of $\beta$, the process approaches deterministic dynamics yielding either ordered or disordered states for smaller or larger $\Delta J$, respectively. We remark that the disordered state with $m = 0$ and $q = 0$ at high $\beta$ (low temperature) does not indicate the spin-glass phase as expected for the equilibrium Ising system (see Supplementary Note 4). We confirmed the non-existence of a spin-glass phase for the asymmetric kinetic SK model by finding that the system decays exponentially in this region (Supplementary Note 5).

The reduction in uncertainty at higher $\beta$ is indicated by the reduction of the conditional entropy (the path entropy) $S_{u|u-1}$ by increasing $\beta$ (Fig. 3a). This figure additionally shows that the conditional entropy decreases slowly with increasing $\beta$ along the critical line of the phase transitions. This means that strong couplings and diverse patterns co-exist along the critical line. In contrast, the time-reversed conditional entropy $S^r_{u|u-1}$ (Fig. 3b) displays opposite dependency on $\beta$ for the broader or narrower coupling distributions. Time-reversed conditional entropy quantifies how surprising the reverse process is under the forward model. With coupling distributions narrower than the critical value $\Delta J^c(\infty)$, the time-reversed conditional entropy diminishes by increasing $\beta$, indicating that the reverse processes takes place with increasingly high probabilities. This is because the spin state is fixated at all up or down under the ferromagnetic-like state for all time, losing temporal asymmetry. In contrast, the reverse process becomes less likely to happen as the dynamics becomes more deterministic by increasing $\beta$ yet remains disordered as long as the coupling distribution is broader than $\Delta J^c(\infty)$. This distinct behavior between the conditional entropy and its time-reversed version found at the wider coupling distributions and high inverse temperatures yields the strong time-asymmetry in this regime.

The entropy production under the steady-state condition quantifies the difference between the conditional and time-reversed conditional entropy. Figure 3c displays the phase diagram of the entropy production for the synchronous Ising model (the asynchronous Ising model has a similar behavior but different scaling, see Supplementary Note 2 or Fig. 4). The entropy production is maximized at the high $\beta$ under the broader coupling distributions, where we find a significant difference between these two conditional entropies. Namely, strong time-asymmetry appears when the dynamics are disordered, nearly deterministic processes. The entropy production increases with $\beta$ if the coupling distribution is wider than $\Delta J^c(\infty)$. In contrast, the entropy production is locally maximized at the critical point (white dashed line) with the coupling distribution being narrower than $\Delta J^c(\infty)$ (see also Fig. 3d). For the narrowly distributed couplings, the process exhibits a paramagnetic-like (randomized or disordered) phase at lower $\beta$ and a ferromagnetic-like (ordered) phase at higher $\beta$ (Fig. 2), neither of which can exhibit adequately asymmetric dynamics in time. Time-asymmetry appears between the ordered and disordered phases,

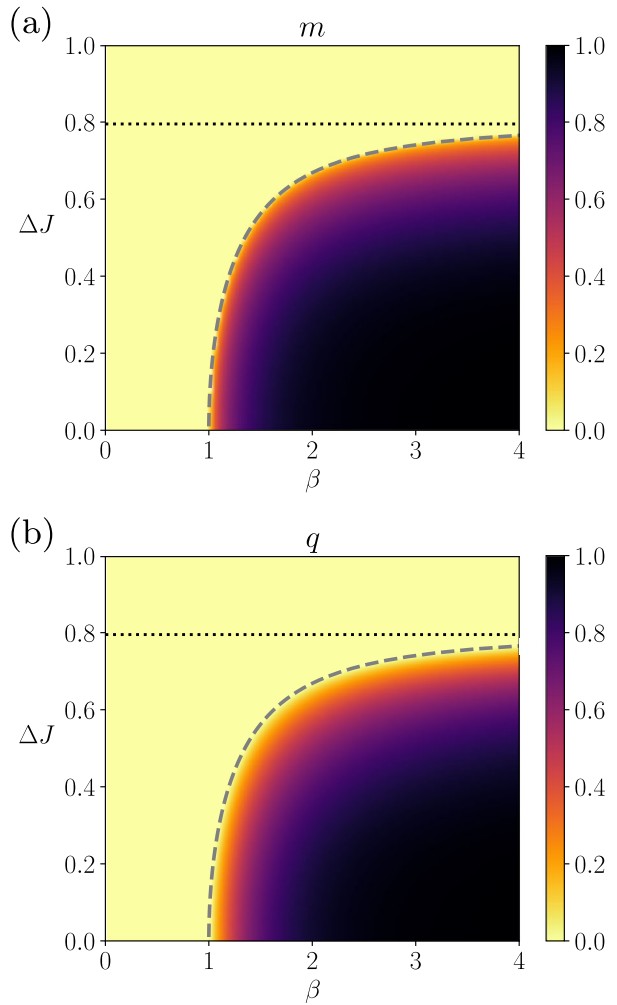

**Fig. 2 | Order parameters of the asymmetric SK model with zero fields under synchronous and asynchronous updates.** The average magnetization $m$ and the average delayed self-coupling $q$ are shown in the phase diagram of the inverse temperature $\beta$ and coupling heterogeneity $\Delta J$ using a model with fixed parameters $J_0 = 1$, $\Delta H = 0$. The dashed line represents the critical line separating ordered and disordered phases. The dotted line represents the critical value at zero temperature ($\beta \to \infty$).

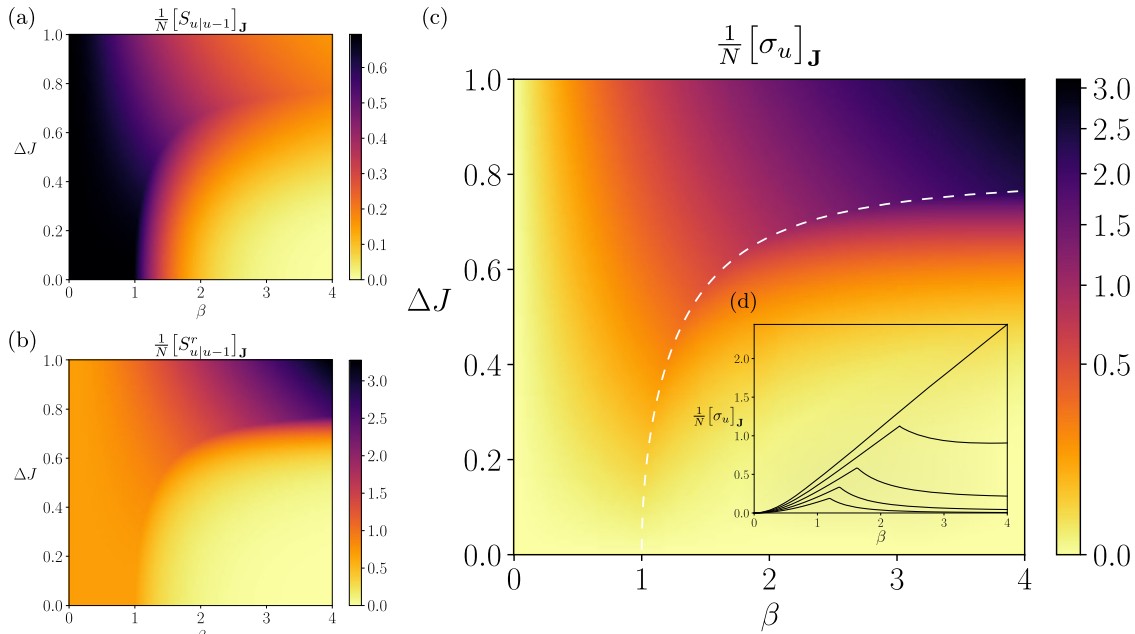

**Fig. 3 | Steady-state entropy rate and entropy production of the asymmetric SK model under synchronous updates. a** The phase diagram of the conditional entropy $[S_{u|u-1}]_J$ (equivalent to the entropy rate) as a function of the inverse temperature $\beta$ and the coupling heterogeneity $\Delta J$. **b** The conditional entropy of the reverse dynamics $[S^r_{u|u-1}]_J$. **c** The entropy production at a steady state. The white dashed line is a critical line for the nonequilibrium phase transitions. **d** (inset) The horizontal sections of the entropy production ($\Delta J = 0.4, 0.5, 0.6, 0.7$, and $0.7950$), showing that it peaks at the critical line. All figures are based on a model with fixed parameters $H_{i,u} = 0$ and $J_0 = 1$.

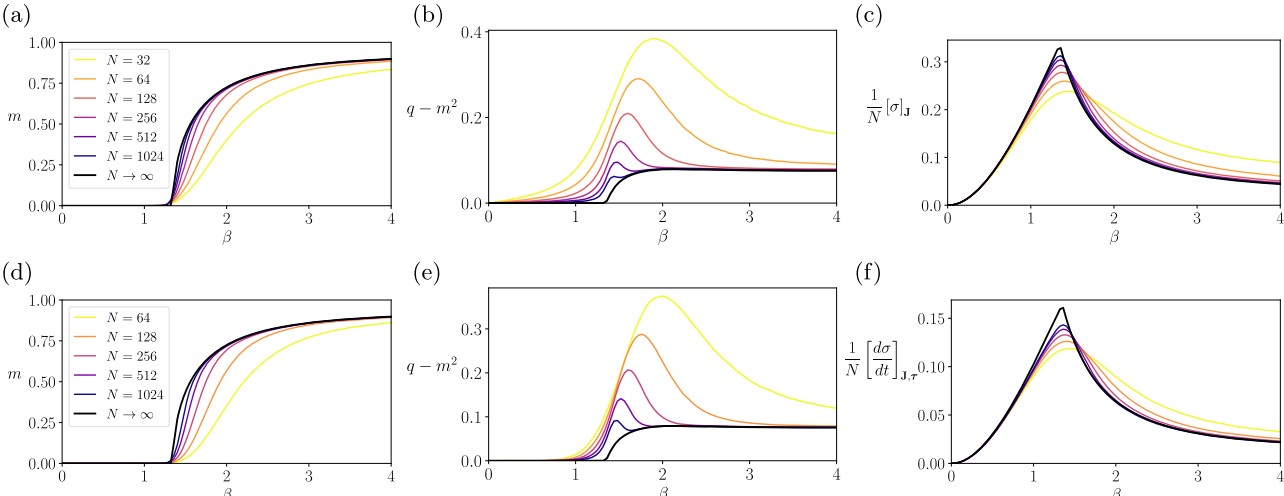

**Fig. 4 | Verification of the exact mean-field solutions by simulating the kinetic Ising systems with synchronous and asynchronous updates.** We repeated $400,000$ simulations of systems under synchronous (top) and asynchronous (bottom) updates with $\Theta_{i,u} = 0$ and $\Delta J = 0.5$: **a, d** Sample estimates of the mean activation rate $\frac{1}{N}\sum_i [\langle s_{i,u}\rangle]_{J,\tau}$ compared with the theoretical order parameter $m$. **b, e** Sample estimates of the average delayed self-covariances $\frac{1}{N}\sum_i [\langle s_{i,u}s_{i,u-d}\rangle]_{J,\tau} -$ $\frac{1}{N}\sum_i [\langle s_{i,u}\rangle]_{J,\tau}\frac{1}{N}\sum_i [\langle s_{i,u-d}\rangle]_{J,\tau}$ ($d = 1$ for the synchronous system and $d = 10N$ for the asynchronous one) computed from samples compared with the theoretical order parameter $q$-$m^2$. **c, f** Sample estimates of the entropy production and entropy production rate (Supplementary Eqs. (S6.6) and (S6.7)) compared with its mean-field value at the thermodynamic limit $\frac{1}{N}[\sigma]_J$, $\frac{1}{N}[\frac{d\sigma}{dt}]_{J,\tau}$ (Eqs. (54) and (58)).

namely at the critical point. As a consequence, the steady-state entropy production can be a measure of the criticality in this regime. However, more importantly, the magnitude of the entropy production is far more significant in the regime of large $\Delta J$ and $\beta$ than near the critical states, due to the strong time-asymmetry caused by the combination of disordered and quasi-deterministic dynamics.

To verify our theoretical predictions for the order parameters and steady-state entropy production, we compared them with the values computed from the sample trajectories by numerically simulating the kinetic SK models (see Supplementary Note 6 for the details). We constructed the kinetic Ising model with parameters $\Theta_{i,u} = 0$ and randomly generated $J_{ij}$ with $\Delta J = 0.5$ and $J_0 = 1$ while changing the inverse temperature $\beta$. We ran simulations of the model for $t = 128$ time steps and repeated the simulation $10^6$ times at each $\beta$. We computed the mean activation rate $\frac{1}{N}\sum_i [\langle s_{i,u}\rangle]_{J,\tau}$, the average delayed self-correlations $\frac{1}{N}\sum_i [\langle s_{i,u}s_{i,u-1}\rangle]_{J,\tau}$, and the normalized entropy production and entropy production rates $\frac{1}{N}[\sigma]_J$, $\frac{1}{N}[\frac{d\sigma}{dt}]_{J,\tau}$ from trajectory and parameter sampling. We used the values at the last time step ($u = t$), where we confirmed that the statistics approached their steady-state values.

Figure 4 compares the theoretical order parameter $m$ and $q$ with the mean activation rate and delayed self-correlations computed from the simulated trajectories for system size $N = 32, \ldots, 1024$ under synchronous (Fig. 4a, b) and asynchronous (Fig. 4d, e) updates. The simulated values approach the theoretical prediction as the size increases, albeit the convergence speed slows down near the critical temperature as it is expected. Similarly, we confirm in Fig. 4c, f that the entropy production from the sample trajectories for synchronous and asynchronous systems converges to the mean-field value at the thermodynamic limit as we increase the system size. Note that entropy production for synchronous updates differs from the entropy rate in the asynchronous update in continuous-time limit due to different values for the delayed correlation term $q$ in Eqs. (54) and (58). These results corroborate our theoretical predictions that the steady-state entropy production peaks at the critical nonequilibrium phase transitions. We further verified by simulations with $\Delta J = 1$ that the steady-state entropy production increases when the significantly heterogeneous systems approach the quasi-deterministic regime (Supplementary Fig. S4).

Finally, knowing the order parameters of the system under the configurational average, we can investigate the structure of the patterns emerging from the dynamics of a sufficiently large but finite system under certain conditions. We calculate the probability $\Omega^{(n)}$ of observing a state $\mathbf{s}_{u+n}$ again for the first time after $n$ steps, starting from the same pattern $\mathbf{s}_u = \mathbf{s}_{u+n}$ (Supplementary Note 7). For zero temperature ($\beta \to \infty$) and synchronous updates, $\Omega^{(n)}$ describes the probability of observing a periodic pattern of length $n$ since transitions become deterministic and can result in periodic patterns. In general, the distribution of these patterns depends on the higher-order correlations between spins across time steps. However, we observe that for the disordered phase ($m = 0$) as well as in the *deep* ordered phase ($m \sim 1$), these correlations disappear for the configurational average in the asynchronous model or in the synchronous model for large $n$ (see Supplementary Fig. S5). In these regions of the phase diagram, we can approximate the probability of observing a pattern $n$ as $\left[\Omega^{(n)}\right] \approx \exp[(1-n)\lambda]$ with $\lambda = \left(\frac{1+m}{2}\right)^N$ (see Supplementary Note 7). The expected length until a repeated pattern is observed is then

$$\sum_n n\left[\Omega^{(n)}\right] \approx \frac{1}{\lambda} = \left(\frac{2}{1+m}\right)^N. \tag{60}$$

At the disordered phase ($m = 0$), the average length of observed patterns exhibits a maximum value, growing exponentially at a rate of $2^N$. In contrast, when the system enters the ordered phase ($m \sim 1$), the growth rate decreases as $m$ increases. In the limit $m = 1$, the system reaches a static equilibrium of average length 1, where the same pattern is repeated indefinitely. These results are consistent with expected dynamics under order-disorder phase transitions. Thus, bringing a quasi-deterministic system to a more stochastic regime by decreasing $\beta$ to the critical value $\beta_c$ (with $\Delta J$ smaller than $\Delta J^c(\infty)$) increases the diversity of irreversible patterns and hence entropy production. However, further reduction of $\beta$ makes the system more random (i.e., less irreversible transitions), leading to a decrease in entropy production. In contrast, the large entropy production found at the disordered phase at large $\beta$ and $\Delta J$ (wider than $\Delta J^c(\infty)$) is caused by diverse oscillatory dynamics whose average pattern length is $2^N$ as in the random dynamics ($\beta = 0$). Adding stochasticity to the dynamics by decreasing $\beta$ in this regime reduces the entropy production monotonically.

### The SK model with uniformly distributed external fields

Next, we apply non-zero external fields to the spins. For simplicity, we will consider the synchronous Ising system with unchanging fields $H_{i,u} = H_i$, assuming a uniform distribution $H_i \sim U(-\Delta H, \Delta H)$. Figure 5a, b shows the $\beta - \Delta J$ phase diagram for the order parameters. With this

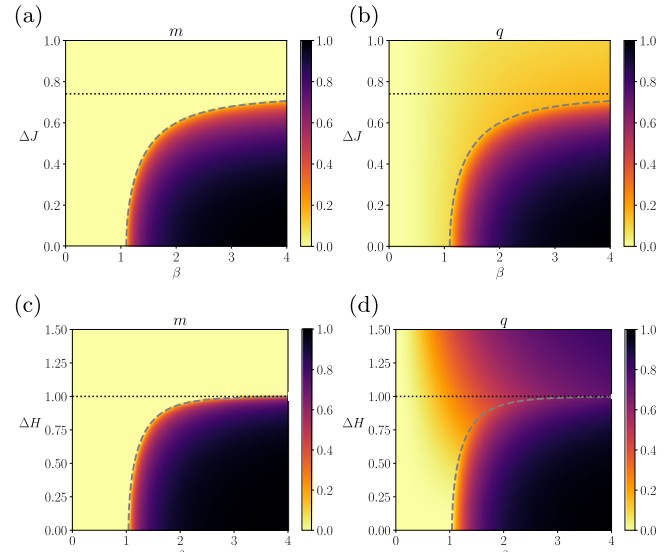

**Fig. 5 | Order parameters of the asymmetric SK model with heterogeneous fields. a, b** The average magnetization $m$ and average delayed self-coupling $q$ are shown as a function of $\Delta J$ and $\beta$. Fixed parameters are $J_0 = 1$, $\Delta H = 0.5$. The dashed line represents the critical line separating ordered and disordered phases. The horizontal dotted line represents the critical $\Delta J$ at zero temperature ($\beta \to \infty$). **c, d** The phase diagram of order parameters as a function of $\Delta H$ and $\beta$. Fixed parameters are $J_0 = 1$ and $\Delta J = 0.2$ and variable $\Delta H$. The dashed line is a critical $\Delta H$ at zero temperature.

change, we observe non-zero correlation $q$ in the area where we previously saw the disordered states ($m = 0$ and $q = 0$, Fig. 2b). Figure 5c, d displays the order parameters as a function of the inverse temperature and $\Delta H$, where we examine the effect of heterogeneity in the external fields while fixing the coupling variability, $\Delta J = 0.2$. The critical line of $\Delta H^c(\beta)$ is obtained in this case as a solution of the following self-consistent equation (Supplementary Eq. (S3.9)):

$$\frac{\Delta H}{J_0} = \int Dz \, \tanh[\beta(\Delta H + \Delta J z)]. \tag{61}$$

Again, as studied in Supplementary Note 3, this critical phase transition corresponds to the mean-field universality class. Since the right-hand side term is less than or equal to 1 regardless of $\beta$ and $\Delta J$, the phase transition occurs only when $\Delta H < J_0$ is satisfied. Intuitively, there is a competition between the dispersion induced by the field diversities $\Delta H$ and the cohesion induced by the mean coupling strength $J_0$. The ordered phase takes place only if $J_0$ counteracts the dispersion induced by the heterogeneity of external fields. More precisely, the critical $\Delta H^c(\beta)$ at the low temperature limit ($\beta \to \infty$) is obtained by solving $\Delta H/J_0 = \int Dz \, \text{sign}[\Delta H + \Delta J z]$. Here we have $\Delta H^c(\infty) = 1$. We observe the phase transition by varying $\beta$ if $\Delta H < \Delta H^c(\infty)$, and no phase transition if $\Delta H > \Delta H^c(\infty)$. Note that $q$ increases monotonically with $\beta$ even for $\Delta H > \Delta H^c(\infty)$ when the distributed fields are introduced.

We now examine the conditional entropy, its reverse, and entropy production for the synchronous system with distributed fields, using the $\beta$ vs. $\Delta H$ phase diagram. Similarly to the observation in the model without fields, the conditional entropy decreases with higher $\beta$ (it becomes more deterministic processes, see Fig. 6a). The time-reversed conditional entropy also decreases with increasing $\beta$ for all $\Delta H$, indicating that the reverse process is more and more likely to happen regardless of $\Delta H$ (Fig. 6b). As seen previously, the time-reversed conditional entropy diminishes under the ferromagnetic-like states ($\Delta H < \Delta H^c(\infty)$). In contrast, we also observe the reduction of the time-reversed conditional entropy at higher $\beta$ for $\Delta H > \Delta H^c(\infty)$. Note that we

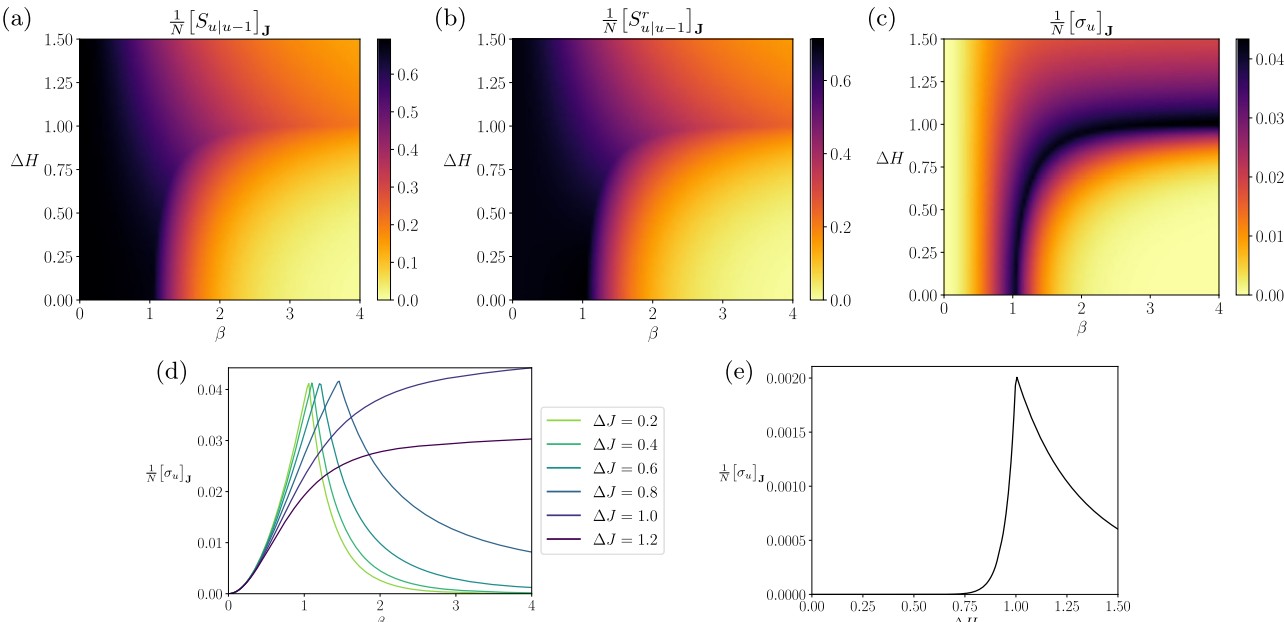

**Fig. 6 | Conditional entropies and entropy production of the asymmetric SK model with heterogeneous fields under synchronous updates. a** The normalized conditional entropy $S_{u|u-1}$ (equivalent to the entropy rate under a steady state). **b** the normalized conditional entropy of the reverse dynamics $S^r_{u|u-1}$. **c** The normalized entropy production at a steady state. **d** Horizontal sections of the entropy production ($\Delta H = 0.2, 0.4, 0.6, 0.8, 1.0,$ and $1.2$). It peaks at the critical line. **e** A vertical section of the entropy production at zero temperature ($\beta \to \infty$). All plots are based on a model with fixed parameters $J_0 = 1, \Delta J = 0.2$ and variable $\Delta H$ and $\beta$.

observe increased correlations $q$ at higher $\beta$ for $\Delta H > \Delta H^c(\infty)$ when we introduce non-zero external fields (see Fig. 5d), which results in a reduction of the reversed entropy similarly to ferromagnetic-like states. Both conditional and time-reversed conditional entropies decrease much slower along the critical line than in other regions, although with different magnitudes. As a result, we see the maximization of the entropy production around critical points more clearly than the $\beta$–$\Delta J$ phase diagram (Fig. 6c, d). Finally, at the zero temperature limit ($\beta \to \infty$), the entropy production peaks at $\Delta H^c(\infty) = 1$ (Fig. 6e).

## Discussion

In this paper, we studied in detail the nonequilibrium thermodynamics of the kinetic, asymmetric SK model for both synchronous and asynchronous dynamics. As expected, the order parameters reveal that the model exhibits order-disorder nonequilibrium phase transitions analogous to the paramagnetic-ferromagnetic phase transitions in the equilibrium Ising model. There are, however, no phase transition akin to a spin-glass (which does not emerge due to coupling asymmetry, as previously reported for continuous-time asymmetric SK models[25]). In addition, we show that the steady-state entropy production is maximized near nonequilibrium phase transition points, being its first derivative discontinuous at these points (Fig. 3d). This result is similar to previously reported critical behavior of the entropy production caused by external stimulation or inertial dynamics (via self-coupling) in homogeneous systems with asynchronous updates by means of naive mean-field approximations or numerical simulations[28–30]. Nevertheless, our result is novel in that it provides the critical behavior of the entropy production caused by asymmetric, heterogeneous couplings using an exact analytical solution for such complex systems. In addition, the studied model displays a region with disordered oscillations in its phase diagram, where the entropy production takes even larger values than in the critical regime. This phase takes place for disordered systems with low entropy rates, i.e., the heterogeneous connections are strong enough to make the dynamics disordered but quasi-deterministic (Fig. 3c, top-right). In contrast, the entropy production does not increase when we increase the heterogeneity of external fields (Fig. 6c).

Taken together, our results indicate that the behavior of entropy production peaking at a critical point is more general than the simple mean-field, homogeneous models, therefore a non-smooth change of the steady-state entropy production (or entropy dissipated to an external reservoir) can be a useful indicator of a number of nonequilibrium phase transitions. At the same time, our results demonstrate that an increase in entropy production does not necessarily mean that the system is approaching a phase transition point. Instead, combining the order parameters, entropy rate, and entropy production yields a more precise picture of the complex systems and their phase transitions.

Typically, solutions of the symmetric (equilibrium) SK model involve the replica trick to calculate the configurational average of the logarithm of the partition function[12]. This method introduces an integer number of replicas of a system for averaging the disorder and then recovers the solution using a continuous number of replicas in the zero limit under the replica symmetry or replica-symmetry breaking ansatz. This treatment forces researchers to check the validity of solutions before reaching correct solutions[16,53]. As an alternative to the replica methods, the path integral methods have been widely used in analyzing the symmetric SK model[54,55]. However, for partially- or fully-symmetric SK models, the path integral method does not give a definite analytical solution but needs to be computed with Monte Carlo approaches[56]. Fortunately, the path integral method derives an exact analytical solution for the case of the fully asymmetric nonequilibrium SK model[25], which we extended to cover synchronous and asynchronous updates, and theoretically underpinned their nonequilibrium properties by deriving the exact solution of the steady-state entropy production and entropy rates of the system.

Nonequilibrium properties of biological and adaptive systems have received the attention of neuroscience and biological science communities. For example, increased entropy production in macroscopic neural activity was suggested as a signature of physically and cognitively demanding tasks[4], conscious activity[5,6] or neuropsychiatric diseases like schizophrenia, bipolar disorders, and ADHD[7]. While it is not easy to contrast their findings based on the coarse-grained analysis of ECoG or fMRI data with the present results, our precise characterization of the entropy production of the prototypical system sheds

light on what kind of behaviors we might expect from these complicated systems. Most importantly, our results indicate two scenarios to increase entropy production by controlling the connection heterogeneity ($\Delta J$) and neuron's nonlinearity ($\beta$). These global changes in the model parameters can be achieved in the brain as gain modulation often mediated by neuromodulators[57]. One scenario to increase entropy production is that the system approaches a critical state as seen in the low $\Delta J$ in Fig. 2 or Fig. 5. The other scenario is to make the system more heterogeneous and sensitive by increasing $\Delta J$ and $\beta$. A significant difference is that the former process maintains stochastic nature while the latter yields quasi-deterministic disorder, as indicated by the high or reduced entropy rate. Therefore, the results suggest that it is crucial to investigate the multiple possibilities of nonequilibrium states to underpin the unconscious (sleep or anesthesia), awake, and engaged states more precisely.

Neuroscientists have often discussed the role of temporal patterns in spiking activities of neurons in computation or in memory consolidation and retrieval. One central topic of the debate is whether neurons, e.g., cortical or hippocampal ones, exhibit precise sequential patterns in a repeated manner[58–61]. Such precise sequences should result in a large entropy production similar to the low-temperature disordered phase of the kinetic Ising system. Alternatively, one may explain a broad range of irreversible temporal patterns, including avalanche dynamics[62,63], by the dynamics near the nonequilibrium phase transitions without the precise sequential structure. As suggested by Eq. (60), the same network that shows simple periodic patterns at zero temperature can retrieve the diverse patterns yielding large entropy production by being poised near the critical phase transition point. The current study highlights the need to dissociate the two scenarios, characterizing different temporally irreversible spiking patterns to understand the distinct roles in neural computation using multiple thermodynamic quantities.

Finally, our analytical solutions offer a benchmark for – the aforementioned and other – methods for estimating thermodynamic quantities. For example, characterizing entropy production from brain imaging data requires methods for coarse-graining the phase diagram[4,5]. The kinetic SK model can serve as a test bench for such methods as it is an analytically tractable system with a well-known phase diagram. Moreover, we can use them to examine both established and novel mean-field theories in estimating the thermodynamic properties of large-scale systems. For example, one can directly fit the Ising model to neuronal spiking data using mean-field methods for finite-size networks, from which one can estimate various thermodynamic quantities of the system[31,64]. Accurately estimating these quantities in large networks gives deeper insights into the nonlinear computations of cortical circuitries. The exact solutions provided here serve to evaluate the accuracy of these approximation methods applied to large-scale networks and provide a benchmark of the thermodynamics quantities of infinitely large networks.

## Data availability
The datasets and code generated in the current study are available in the GitHub repository, https://github.com/MiguelAguilera/asymmetric-SK-model.

## Code availability
The datasets and code generated in the current study are available in the GitHub repository, https://github.com/MiguelAguilera/asymmetric-SK-model.

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

## Acknowledgements

MA was partially funded by the European Union's Horizon 2020 research and innovation programme under the Marie Skłodowska-Curie grant agreement No 892715 and a Junior Leader fellowship from "la Caixa" Foundation (ID 100010434, code LCF/BQ/PI23/11970024). HS is supported by JSPS KAKENHI Grant Number JP 20K11709, 21H05246, the National Institutes of Natural Sciences (NINS Program No. 01112005, 01112102), and New Energy and Industrial Technology Development Organization (NEDO), Japan.

## Author contributions

M.A. and H.S. designed and reviewed the research; M.A. contributed the analytical results. M.A. and M.I. contributed the numerical results; M.A., M.I. and H.S. contributed the theoretical review and wrote the paper.

## Competing interests

The authors declare no competing interests.
