## [Peer Review File · Nature Communications]

REVIEWER COMMENTS

Reviewer #1 (Remarks to the Author):

Authors propose an interesting paper about the statistical mechanics and phase transitions of a nonequilibrium version of the Sherrington-Kirkpatrick model. By means of a general analysis, based on generating functional for Markovian chains, analytical expressions for an asymmetric version SK were obtained.

The manuscript is interesting and contributes to realm of nonequilibrium phase transitions by means of a genuinely nonequilibrium quantity: the entropy production. However, before taking my decision about accepting or not the present manuscript, authors have to answer properly my questions:

1. Please define properly the SK in terms of transition rates and explain why the analyzed version evolves to a nonequilibrium steady state regime.
2. Please explain in more details how Eq. (18) were introduced.
3. Expression for conditional probability given by Eq. (18) is not clear for me. Please explain how it was deduced.
4. What are critical exponents from critical behaviors from Figs. 2-4? Are they classical as calculations derived in appendix B for $N \rightarrow \infty$? How they compare with critical behavior for the equilibrium version? Please clarify these points.

5. Again respect to the critical behavior, authors claim that entropy production peaks at the critical point and it would properly locate the critical point. However, inspection of figures 3, 5 together expressions from appendix B suggest that critical exponents are indeed classical, implying that findings are similar to those band from Refs. [30-32]. In other words, all entropy production curves exhibit a kink at the criticality, consistent with its derivative (respecting the control parameter) scaling with the classical exponent $\alpha=0$. Indeed, Ref. [31] explains why the entropy production is peaked at the criticality for Ising like models.

6. Despite the analyticity of results, it should be remarkable authors perform some numerical simulations in order to validate all analytical findings. In other words, it should be interesting authors compare some exact entropy production curves with numerical simulation ones.

Minor points:

7. The discussion about the interplay between random and deterministic transitions (end of 8th page) sounds very confuse to me.

8. From figure 4, it seems that the phase diagrams ΔH versus β and ΔJ versus β system behave similarly. Is it a reason for such behavior?

9. Please show graphics from Fig. 5 (panel d) by using distinct colors.

The caption description does not provide me to discern the corresponding parameter for each curve.

Reviewer #2 (Remarks to the Author):

In this paper, the authors develop a field-theoretic formalism for calculating entropy production in irreversible stochastic systems and apply it to the asymmetric SK model. The model is exactly soluble by their method. They find novel results about the temperature- and disorder strength-dependence of the entropy production that have generic implications for other irreversible systems.

Non-equilibrium statistical mechanics has been an active field for a while, and field-theoretic formalisms (e.g. Martin-Siggia-Rose) have been developed and applied to a variety of problems. However, until now there has been no method for systematically calculating the quantity that defines and measures the irreversibility of a system: the entropy production. As such, the paper therefore represents a major step forward in the field.

The formal mathematical development extends, in a nontrivial and important way, some methods used in the past for this model and other nonequilibrium systems. People familiar with that work will recognize the techniques and be able to read the paper easily. For non-specialists it will require more effort, but the description here is careful and systematic and should be clear to motivated readers who want to learn the method. For a more general readership, the thinking behind the math should be evident and the results should be clear.

Other work has led to a general idea that entropy production is maximal near 2nd-order phase transitions. The results here show that this is only part of the story -- it can also be high at low temperatures when, as in the model studied here, the intrinsic temporal asymmetry of the dynamics lead to highly chaotic, nearly deterministic behaviour.

The model studied here is of "mean-field" type (weak infinite-range interactions), and this feature is essential to its solubility. However, in more realistic models mean field theory can be used as the starting point for more complete treatments (loop expansions, etc.), so this paper can also lay the groundwork for calculations in a much more general class of problems.

To summarize: while the paper's concrete results are limited and restricted to an artificial model designed so as to be solvable using mean field theory, it is extremely valuable for setting up a theoretical machinery with which one will be able to investigate the statistical dynamics of non-equilibrium systems in much greater depth than was previously possible. Without question, it should be published.

I did find one typo: In the last sentence of Sect. V the zero-temperature limit in parentheses should be $\beta \rightarrow \infty$, not $\beta \rightarrow 0$.

Reviewer #3 (Remarks to the Author):

The authors study the dynamics of the asymmetric SK model with a path integral method, finding analytical equations describing the order parameters: the magnetization $m(t)$ and the self-correlation at different times $q(t, t')$. They also characterize the entropy production of the process.

They find no-spin glass phase at difference with the symmetric SK model, and they find that entropy production is locally maximized on the critical line, but it can be globally maximized far from criticality and this is an interesting finding that could be useful in characterizing other out-of-equilibrium systems.

Precedent works on the subject are not cited or discussed in a proper way. I list here some examples:

- The idea to study the asymmetric spin-glass model is not new. In fact the dynamics of the fully asymmetric SK model was already analytically studied in ref. [51] (Crisanti/Sompolinsky) and the conclusion that asymmetry destroys the spin-glass phase was already known. The authors should discuss the paper [51] saying explicitly what was already done and which are the differences with their work.

- In the conclusion one reads "Typically, solutions of the symmetric (equilibrium) SK model involve the replica trick to calculate the configurational average of the logarithm of the partition function [12]. This method involves contradictory assumptions by introducing an integer number of replicas of a system for averaging disorder and then recovering the solution using a continuous number of

replicas in the zero limit. This apparent contradiction is not formally resolved, and the method does not always result in the exact solutions, so one must introduce additional assumptions such as replica symmetry-breaking. The path integral method is free from these assumptions, although it can be less flexible than the replica method.” In my opinion this sentence is not fairly written. In fact the problem of the replica-symmetric solution not being the correct solution of the problem is not caused by the contradictory assumption of the number of replica being integer and then analytically continued to zero. Replica-symmetry is not correct because the system breaks it spontaneously. At this point one breaks the replica symmetry, but again this is not an assumption but what is physically happening: the phase space is divided in many equilibrium states and replicas are just proxies for them (just as a ferromagnet breaks the spin-inversion symmetry). Indeed the replica symmetry breaking scheme of ref. [16] has been proved mathematically to be exact (see the works of Guerra/Talagrand/Panchenko).

- In the conclusion one reads: “For the case of the symmetric SK model, the path integral method does not give a definite analytical solution but needs to be computed with Monte Carlo approaches [52]”. Ref. 52 does not seem to be the right one to be cited. Indeed, there was a lot of work on dynamics of symmetric SK model where analytical solutions can be obtained through path-integral methods in the high temperature phase (Sompolinsky-Zippelius PRB 1982) and in the low-temperature out-of-equilibrium regime with a proper scaling of the times (Cugliandolo, Kurchan. Journal of Physics A 1994).

I then list two major points about the support for the conclusions that the authors should address:

- Through the use of a path integral method the authors derive analytical equations for the evolution of the order parameters of the model. However, as usually done after involved analytical calculations, theoretical results should be checked by numerical simulations (like Monte-Carlo) of the dynamics, showing that the numerical data are well reproduced by the analytical equations. The paper lacks for this fundamental step. I suggest to look to the numerical behavior of $m(t)$ and $q(t,t')$ and check if their evolution is described by the obtained analytical equations.

-The dynamics of the asymmetric model is studied under parallel update of all the N spins. However it is known that in disordered systems parallel or sequential update can lead to very different behavior. When checking for the numerical verifications of the analytical results, suggested in the previous point, one should look both to parallel and sequential updates and see if there are differences in the results.

Some technical points (there could be other typos thus I suggest a careful check of all the equations):

-After eq. (20), “ $i=1,\dots,N$ and $u=1,\dots,t$ ” : add range for σ

- eq. (26), first term of second line: replace $()$ with $\langle \rangle_g$

- after eq. (33) a delta function is introduced (as an integral in $\hat{\theta}$) to define θ . It seems to me that, in the same way, a delta function should be introduced to define ϑ (as an integral in $\hat{\vartheta}$). This delta function, the integrals in ϑ and $\hat{\vartheta}$ are missing in eqs. (34) and (A6)

- eq (44) first line, last term: put $i,u-1$ as subscript

- caption of Fig. 4: In the caption: substitute (b,c) with (c,d) . “Fixed parameters are $J_0 = 1$ and $\Delta J = 0.2$ and variable.” Add ΔH .

- In the main text the steady state values for m and q are mentioned without being defined. They have been defined in the Supplementary Material (after eq. A63, one reads “since every $q_{u,v}$ depends only on m_u and m_v , all $q_{u,v}$ converges to the same solution, q , under the steady state.”) This definition should be added also to the main text.

- eq A20: substitute the second $1/2(M^+)^2$ with M^-

-eq A21: substitute the second $1/2(Q^+)^2$ with Q^-

-eq A25: M and Q should be substituted by M^* and Q^*

-first line of pag 8: substitute $\hat{\vartheta}$ with $\hat{\theta}$

- eq. A56: $g_{j,v}$ should be replaced by $g_{i,v}$

-on Fig. S2: I think the y-label is not m , but $m(t)-m_{\text{steadystate}}$. Moreover, in the caption it should be mentioned the difference between (a) and (b) (I think it is only the log-scale)

- Bibliographic references are wrong in the Supplementary materials: for example at page 9 ref. [49] should be replaced by ref. [51]. Ref. [60] at page 12 is not present in the bibliography

We appreciate the reviewers' critical comments, which have allowed us to substantially improve the manuscript. Before responding to each review in detail, here we provide a summary of major changes in the document:

1. Following the suggestion of reviewer #1, we inserted a new subsection about the nonequilibrium steady state. We described conditions to achieve the steady state under equilibrium or nonequilibrium conditions before introducing the steady-state entropy production.
2. Following the suggestion of reviewer #3, we found that we could generalize our results on the synchronous Ising model to the asynchronous kinetic Ising model. We inserted a new Supplementary Note 2, where we detailed the generalization. We also modified the manuscript to emphasize that our results apply to both cases.
3. Following the suggestions by reviewers #1 and #3, we confirmed our theoretical prediction by numerically simulating the kinetic Ising systems for both synchronous and asynchronous updates (See the new Fig. 4, Fig.S5 and Supplementary Note 6). Due to computational time constraints, so far we have simulated networks of up to 512 neurons, but, we can add the simulations up to sizes of at least 1024 before publication.

Thank you for kindly revising our manuscript. We hope that our responses are satisfactory.

Sincerely,
Miguel Aguilera

Reviewers' comments: black
Our responses: blue

REVIEWER COMMENTS

Reviewer #1 (Remarks to the Author):

Authors propose an interesting paper about the statistical mechanics and phase transitions of a nonequilibrium version of the Sherrington-Kirkpatrick model. By means of a general analysis, based on generating functional for Markovian chains, analytical expressions for an asymmetric version SK were obtained.

The manuscript is interesting and contributes to realm of nonequilibrium phase transitions by means of a genuinely nonequilibrium quantity: the entropy production. However, before taking my decision about accepting or not the present manuscript, authors have to answer properly my questions:

1. Please define properly the SK in terms of transition rates and explain why the analyzed version evolves to a nonequilibrium steady state regime.

We inserted a new section in the revised manuscript, 'nonequilibrium steady state,' before introducing the entropy production. Here we defined the transition rates for the discrete and continuous-time systems. We clearly stated the conditions that the Markov process reaches the steady state (irreducibility and aperiodicity). The maximum entropy Markov process meets these criteria. Given this, we formulated the equilibrium and nonequilibrium steady states.

We also introduced the dissipation function in the entropy production section because it directly assesses the violation of the detailed balance that defines if the system is in the equilibrium or nonequilibrium state in general. However, since the (total) entropy production (that we defined in the previous manuscript) is more widely used, and the dissipation function and (total) entropy production are equivalent under the steady state or continuous-time limit, we kept the description on the (total) entropy production as well.

2. Please explain in more details how Eq. (18) were introduced.

We rewrote the introductory sentences for the generating functional, emphasizing that Eq.~18 (in the previous manuscript) was designed to retrieve the expectation of the statistics we are interested in, where the expectation was taken over the ensembles of the trajectories.

It now reads as follows (page 4).

"In equilibrium systems, the partition function retrieves its statistical moments. A nonequilibrium equivalent function is a generating functional or dynamical partition function. To obtain not only the statistics averaged over trajectories, but also the forward/reversed conditional entropies (Eqs.~\ref{eq:conditional-entropy}, \ref{eq:reverse_conditional-entropy}), we define a generating functional:"

3. Expression for conditional probability given by Eq. (18) is not clear for me. Please explain how it was deduced.

The previous equation of the generating functional (see below) was not intuitive.

$$Z_t(\mathbf{g}) = \sum_{\mathbf{s}_{0:t}} \exp \left[- \sum_u \epsilon(\mathbf{s}_u | \mathbf{s}_{u-1}) + \Gamma(\mathbf{g}, \mathbf{s}_{0:t}) \right] p(\mathbf{s}_0), \quad (18)$$

The epsilon in Eq.18 is - log of the conditional (transition) probability. We simplified the equation as follows to make it clear that it considers the ensemble average:

$$Z_t(\mathbf{g}) = \sum_{\mathbf{s}_{0:t}} p(\mathbf{s}_{0:t}) \exp \left[\Gamma(\mathbf{g}, \mathbf{s}_{0:t}) \right], \quad (24)$$

4. What are critical exponents from critical behaviors from Figs. 2-4?

Are they classical as calculations derived in appendix B for $N \rightarrow \infty$?

How they compare with critical behavior for the equilibrium version? Please clarify these

points.

Yes, they are classical. Figs. 2-3 are the case without the external fields, and Fig.4 is the case with the external field. Regardless of the presence of the field, we showed in Supplementary Note 3 (previously Appendix B) that the order parameters show the classical scaling exponent of the mean-field universality class. We have clarified this in the text in the results section (below Eqs 52, 54).

We have also clarified in the main text and in the last paragraph of Supplementary Note 3 that these exponents (mean-field universality class) are also found in the order-disorder phase transition of the equilibrium SK model. We also note that these exponents are different in the spin glass phase, not present in the asymmetric model.

5. Again respect to the critical behavior, authors claim that entropy production peaks at the critical point and it would properly locate the critical point. However, inspection of figures 3, 5 together expressions from appendix B suggest that critical exponents are indeed classical, implying that findings are similar to those band from Refs. [30-32]. In other words, all entropy production curves exhibit a kink at the criticality, consistent with its derivative (respecting the control parameter) scaling with the classical exponent $\alpha=0$. Indeed, Ref. [31] explains why the entropy production is peaked at the criticality for Ising like models.

Indeed, as the reviewer points out, the critical exponents are classical, and our findings are similar to those references mentioned in the paper. Still, we note two important differences between our results and the references mentioned.

The first difference is that our results are exact and not the result of simulations or approximations. For example, the results of [31] are either simulations or the result of a naive mean field approximation, assuming an independent probability distribution. In contrast, our independent but history-dependent mean-field solutions obtained by the configurational average include the effect of heterogeneous couplings.

The second difference is the mechanisms inducing a non-zero entropy production. The results in [30, 31, 32] describe homogeneous and symmetrically connected systems (e.g., the majority voter model or an Ising-like lattice). In [30], entropy production arises from the effect of an external influence bringing the system out of equilibrium by forcing transient dynamics. In [31, 32], entropy production arises only from inertia terms (i.e., self-couplings) under asynchronous updates. In contrast, in our paper, entropy production arises from a complicated web of asymmetric, heterogeneous couplings that can give rise, for example, to complex endogenous oscillations in individual trajectories. This behavior may provide a clue on the mechanisms for generating rhythmic activity without external stimulation and may explain the dynamics found in various biorhythms, such as default-mode brain networks.

Hence, we argue that the behavior of entropy production peaking at a critical point is likely to be more general than the simple mean-field, homogeneous models. Even for the more complex system, our approach gives an exact solution of the entropy production, which is new in the literature, and describes the behaviour of large entropy production for disordered,

chaotic-like oscillations (top right region of the phase space).

We clarified these aspects in the paper in the first paragraph of the discussion section (see below), referencing again the references below and explaining the relation with our study.

"This result is similar to the previously reported critical behavior of the entropy production caused by external stimulation or inertial dynamics (via self-coupling) in homogeneous systems with asynchronous updates by means of naive mean-field approximations or numerical simulations \cite{zhangCriticalBehaviorEntropy2016a,noaEntropyProductionTool2019a,crochik2005entropy}. Still, our result is novel in that it provides the critical behavior of the entropy production caused by asymmetric, heterogeneous couplings using an exact analytical solution for such complex systems. "

6. Despite the analyticity of results, it should be remarkable authors perform some numerical simulations in order to validate all analytical findings. In other words, it should be interesting authors compare some exact entropy production curves with numerical simulation ones.

Following the reviewer's suggestion, we performed numerical simulations of the kinetic Ising systems. We computed the average activation rate, the average delayed self-correlations, and the entropy production from the sampling trajectories to compare with the theoretical values. Please see Fig.4 and two inserted paragraphs in Results for details. We confirmed that our path-integral approach accurately describes the simulated results of both systems with synchronous and asynchronous updates.

Minor points:

7. The discussion about the interplay between random and deterministic transitions (end of 8th page) sounds very confuse to me.

We agree that the interplay was not an accurate description of the process but rather our interpretation (possibly misleading). To describe only the results, we removed the sentence "where the process balances the random and deterministic state transitions" from the paragraph. We also removed a similar sentence at the end of the analysis of periodic patterns.

8. From figure 4, it seems that the phase diagrams ΔH versus β and ΔJ versus β system behave similarly. Is it a reason for such behavior?

Both ΔH and ΔJ are measures of variability in the model parameters. They introduce a disorder that makes a transition to an ordered/polarized state more difficult because stronger couplings are required to achieve the transition. Thus, we think they drive a similar phase transition behavior when the disorder driven by ΔH and ΔJ overcomes the order induced by the J_0 parameter.

9. Please shows graphics from Fig. 5 (panel d) by using distinct colors. The caption

description does not provide me to discern the corresponding parameter for each curve.

We have used different colors for each ΔH and added the labels for them for better visibility. We have updated Fig. 3(d) similarly.

Reviewer #2 (Remarks to the Author):

In this paper, the authors develop a field-theoretic formalism for calculating entropy production in irreversible stochastic systems and apply it to the asymmetric SK model. The model is exactly soluble by their method. They find novel results about the temperature- and disorder strength-dependence of the entropy production that have generic implications for other irreversible systems.

Non-equilibrium statistical mechanics has been an active field for a while, and field-theoretic formalisms (e.g. Martin-Siggia-Rose) have been developed and applied to a variety of problems. However, until now there has been no method for systematically calculating the quantity that defines and measures the irreversibility of a system: the entropy production. As such, the paper therefore represents a major step forward in the field.

The formal mathematical development extends, in a nontrivial and important way, some methods used in the past for this model and other nonequilibrium systems. People familiar with that work will recognize the techniques and be able to read the paper easily. For non-specialists it will require more effort, but the description here is careful and systematic and should be clear to motivated readers who want to learn the method. For a more general readership, the thinking behind the math should be evident and the results should be clear.

Other work has led to a general idea that entropy production is maximal near 2nd-order phase transitions. The results here show that this is only part of the story -- it can also be high at low temperatures when, as in the model studied here, the intrinsic temporal asymmetry of the dynamics lead to highly chaotic, nearly deterministic behaviour.

The model studied here is of "mean-field" type (weak infinite-range interactions), and this feature is essential to its solubility. However, in more realistic models mean field theory can be used as the starting point for more complete treatments (loop expansions, etc.), so this paper can also lay the groundwork for calculations in a much more general class of problems.

To summarize: while the paper's concrete results are limited and restricted to an artificial model designed so as to be solvable using mean field theory, it is extremely valuable for setting up a theoretical machinery with which one will be able to investigate the statistical dynamics of non-equilibrium systems in much greater depth than was previously possible. Without question, it should be published.

We are thankful for the reviewer's amiable comments and the succinct description of the significance of our results in a larger context. We integrated the comment about how our results could lay the groundwork for a more general class of problems in the discussion.

I did find one typo: In the last sentence of Sect. V the zero-temperature limit in parentheses should be $\beta \rightarrow \infty$, not $\beta \rightarrow 0$.

We corrected it.

Reviewer #3 (Remarks to the Author):

The authors study the dynamics of the asymmetric SK model with a path integral method, finding analytical equations describing the order parameters: the magnetization $m(t)$ and the self-correlation at different times $q(t, t')$. They also characterize the entropy production of the process.

They find no-spin glass phase at difference with the symmetric SK model, and they find that entropy production is locally maximized on the critical line, but it can be globally maximized far from criticality and this is an interesting finding that could be useful in characterizing other out-of-equilibrium systems.

Precedent works on the subject are not cited or discussed in a proper way. I list here some examples:

- The idea to study the asymmetric spin-glass model is not new. In fact the dynamics of the fully asymmetric SK model was already analytically studied in ref. [51] (Crisanti/Sompolinsky) and the conclusion that asymmetry destroys the spin-glass phase was already known. The authors should discuss the paper [51] saying explicitly what was already done and which are the differences with their work.

We agree with the reviewer that we should have clarified that [51] had already shown the absence of the spin-glass phase, although we mentioned it in the Results section of the previous version. Accordingly, we revised the manuscript as follows:

- Reference [24] (previous 51) and [23] had already solved the fully asymmetric case showing the absence of a spin-glass phase:

"Previous studies using this method \cite{crisanti1987dynamics, crisanti1988dynamics} have shown that the asymmetric kinetic Ising model with asynchronous updates does not have a spin glass phase. In this manuscript, we will extend the generating functional path integral method to confirm this result in both cases of synchronous and asynchronous updates and further retrieve an exact solution of the entropy production of the system."

- We added a citation to a sentence in the initial paragraph of the discussion:

"There are, however, no dynamics akin to the spin-glass phase (which cannot emerge due to coupling asymmetry, as previously reported for continuous-time asymmetric SK systems in \cite{crisanti1988dynamics})."

- We also revised the following sentence in the Result section.

From

"We confirmed the non-existence of a spin-glass phase by finding that the system decays

exponentially in this region (Appendix~D)."

to

"We confirmed the non-existence of a spin-glass phase for the asymmetric kinetic SK model by finding that the system decays exponentially in this region (Supplementary Note~4)."

- We removed the following sentence in abstract:

"The order parameters reveal order-disorder nonequilibrium phase transitions as found in equilibrium systems but no dynamics akin to the spin-glass phase. "

- In the conclusion one reads "Typically, solutions of the symmetric (equilibrium) SK model involve the replica trick to calculate the configurational average of the logarithm of the partition function [12]. This method involves contradictory assumptions by introducing an integer number of replicas of a system for averaging disorder and then recovering the solution using a continuous number of replicas in the zero limit. This apparent contradiction is not formally resolved, and the method does not always result in the exact solutions, so one must introduce additional assumptions such as replica symmetry-breaking. The path integral method is free from these assumptions, although it can be less flexible than the replica method." In my opinion this sentence is not fairly written. In fact the problem of the replica-symmetric solution not being the correct solution of the problem is not caused by the contradictory assumption of the number of replica being integer and then analytically continued to zero. Replica-symmetry is not correct because the system breaks it spontaneously. At this point one breaks the replica symmetry, but again this is not an assumption but what is physically happening: the phase space is divided in many equilibrium states and replicas are just proxies for them (just as a ferromagnet breaks the spin-inversion symmetry). Indeed the replica symmetry breaking scheme of ref. [16] has been proved mathematically to be exact (see the works of Guerra/Talagrand/Panchenko).

We thank the reviewer for pointing this out. Following the reviewer's suggestion, we refrained from using the word "assumption" for the analytic continuation of the replica number. Instead, we introduced the word ansatz of the replica symmetry and replica-symmetry breaking. We stated that the replica method requires additional procedures for checking the validity of solutions, such as positivity of the entropy at low temperatures, under these ansatz before reaching the exact and physically relevant solutions (i.e., the Parisi solution). We here argued that the path-integral approach for the fully asymmetric model does not require these cumbersome steps.

"Typically, solutions of the symmetric (equilibrium) SK model involve the replica trick to calculate the configurational average of the logarithm of the partition function \cite{nishimori_statistical_2001}. This method introduces an integer number of replicas of a system for averaging the disorder and then recovers the solution using a continuous number of replicas in the zero limit under the replica symmetry or replica-symmetry breaking ansatz. This treatment imposes researchers to check the validity of solutions before reaching correct solutions \cite{parisi1979toward,parisi1980sequence}. The path integral method applied to the fully asymmetric model is free from such additional steps and directly provides the exact solutions, although it can be less flexible than the replica method."

- In the conclusion one reads: "For the case of the symmetric SK model, the path integral method does not give a definite analytical solution but needs to be computed with Monte Carlo approaches [52]". Ref. 52 does not seem to be the right one to be cited. Indeed, there was a lot of work on dynamics of symmetric SK model where analytical solutions can be obtained through path-integral methods in the high temperature phase (Sompolinsky-Zippelius PRB 1982) and in the low-temperature out-of-equilibrium regime with a proper scaling of the times (Cugliandolo, Kurchan. Journal of Physics A 1994).

Thanks for noting this. This was a typo, since we meant the case of the "partially asymmetric SK model" (as opposed to fully asymmetric), which is studied in Ref [52]

Since the reviewer kindly pointed out the studies on the path integral approach to the symmetric SK model, we revised the above sentence to include the information about the symmetric, fully asymmetric, and partially asymmetric SK model as follows:

"The path integral methods have been widely used in analyzing the symmetric SK model \cite{sompolinsky1982relaxational}. Similarly, one can apply this method to study the fully asymmetric SK model by considering the configurational average \cite{crisanti1988dynamics}. However, for the case of the partially asymmetric SK model, the path integral method does not give a definite analytical solution but needs to be computed with Monte Carlo approaches \cite{eissfeller1994mean}."

I then list two major points about the support for the conclusions that the authors should address:

- Through the use of a path integral method the authors derive analytical equations for the evolution of the order parameters of the model. However, as usually done after involved analytical calculations, theoretical results should be checked by numerical simulations (like Monte-Carlo) of the dynamics, showing that the numerical data are well reproduced by the analytical equations. The paper lacks for this fundamental step. I suggest to look to the numerical behavior of $m(t)$ and $q(t,t')$ and check if their evolution is described by the obtained analytical equations.

We performed numerical simulations as suggested. We computed $m(t)$ and $q(t,t')$ but also the entropy production from the sampling trajectories and compared them with the theoretical values. Please see Fig.4 and the corresponding paragraphs in Results for details. We confirmed that our path-integral approach accurately describes the simulated results. Please note that we verified our solutions for both asynchronous and synchronous updates, the latter of which was introduced in response to the reviewer's comment below.

-The dynamics of the asymmetric model is studied under parallel update of all the N spins. However it is known that in disordered systems parallel or sequential update can lead to very different behavior. When checking for the numerical verifications of the analytical results, suggested in the previous point, one should look both to parallel and sequential updates and see if there are differences in the results.

In the revised manuscript, we added a new appendix, Supplementary Note 2, in which we

developed theoretical results for the asynchronous kinetic Ising model. We demonstrated that the asynchronous update is realized by adding augmented random binary variables within the field parameters of each spin, and show that it results in a continuous-time master equation for the kinetic Ising system in a proper limit. Thus, these results can be seen as a generalization of our path-integral theory, covering both synchronous and asynchronous dynamics.

In these new theoretical results, we found that the values of the order parameters $m(t)$ and $q(t',t)$ (assuming $t' \gg t$) at the steady state under the asynchronous model are the same as in the synchronous model. We confirmed this prediction by numerical results (Fig. 4d,e). Further, we found that the entropy production has a similar analytical expression, which depends on the delayed self-correlation. However, we found that the delayed self-correlation takes different values in the synchronous and asynchronous cases, leading to a different scaling of the entropy production, although the functional form is quite similar. We confirmed that the theoretical results in the asymmetric model matched numerical predictions (Fig. 4f).

We thank the reviewer for this comment because we could significantly expand the scope of our theory. We revised the entire manuscript to cover the cases for both synchronous and asynchronous updates.

Some technical points (there could be other typos thus I suggest a careful check of all the equations):

We wish to profusely thank the reviewer for the detailed review of our appendix material. See below the corrections we performed

-After eq. (20), “ $i=1,\dots,N$ and $u=1,\dots,t$ ” : add range for σ

Here σ is just a label to denote the parameter to retrieve the entropy production (σ), as opposed to the parameters $g_{\{i,u\}}$ retrieving spin statistics. I.e., σ does not take different values. To avoid confusion, we changed the notation. We have $g_{\{i,u\}}$ for spins, and $g_{\{u\}^S}$ and $g_{\{u\}^R}$ to retrieve the forward and reverse entropy rates .

- eq. (26), first term of second line: replace $()$ with $\langle \rangle_g$

Corrected

- after eq. (33) a delta function is introduced (as an integral in $\hat{\theta}$) to define θ . It seems to me that, in the same way, a delta function should be introduced to define ϑ (as an integral in $\hat{\vartheta}$). This delta function, the integrals in ϑ and $\hat{\vartheta}$ are missing in eqs. (34) and (A6)

Interestingly, the reverse fields can be directly defined from forward fields $h_{\{i,u\}^r} = h_{\{i,u+1\}} + (H_{\{i,u\}} - H_{\{i,u+1\}})$.

Similarly, knowing θ , we can define $\vartheta_{\{i,u\}} = \theta_{\{i,u+1\}} + \beta(H_{\{i,u\}} - H_{\{i,u+1\}})$, therefore we can avoid introducing an additional delta integral.

We have clarified this in the manuscript.

- eq (44) first line, last term: put $i, u-1$ as subscript

Corrected

- caption of Fig. 4: In the caption: substitute (b,c) with (c,d). "Fixed parameters are $J_0 = 1$ and $\Delta J = 0.2$ and variable." Add ΔH .

Corrected

- In the main text the steady state values for m and q are mentioned without being defined. They have been defined in the Supplementary Material (after eq. A63, one reads "since every $q_{u,v}$ depends only on m_u and m_v , all $q_{u,v}$ converges to the same solution, q , under the steady state.") This definition should be added also to the main text.

We introduced the definition of m and q as steady-state solutions of m_u and $q_{\{u,v\}}$ in the main text of the revised manuscript. We also mentioned that these values are independent of u or $|u-v|$ in the symmetric Ising system. Furthermore, we indicate that (as we show in the new Supplementary Note 2) the order parameter for the systems with asynchronous updates in continuous time limit converges to the same value in the steady state under an appropriate condition ($u \gg v$ for $q_{\{u,v\}}$), so we used the same notation (m and q) for the asynchronous systems.

- eq A20: substitute the second $1/2(M^+)^2$ with M^-

Corrected

-eq A21: substitute the second $1/2(Q^+)^2$ with Q^-

Corrected

-eq A25: M and Q should be substituted by M^* and Q^*

Corrected

-first line of pag 8: substitute $\hat{\vartheta}$ with $\hat{\theta}$

Corrected

- eq. A56: $g_{j,v}$ should be replaced by $g_{i,v}$

Corrected

-on Fig. S2: I think the y-label is not m , but $m(t)-m_{\text{steadystate}}$. Moreover, in the caption it should be mentioned the difference between (a) and (b) (I think it is only the log-scale)

Corrected.

- Bibliographic references are wrong in the Supplementary materials: for example at page 9

ref. [49] should be replaced by ref. [51]. Ref. [60] at page 12 is not present in the bibliography

Corrected

Reviewer #2 (Remarks to the Author):

The authors have made extensive changes in response to the other reviewers' comments, notably the inclusion of simulation results that confirm the analytic calculations and (what is more important, in my opinion) the extension of the theory to asynchronous dynamics.

While I leave it to the other reviewers to judge whether the comments and questions in their first reports have now been answered completely satisfactorily, I think these revisions strengthen the already strong case for publishing this paper.

Reviewer #3 (Remarks to the Author):

The authors added numerical simulations to corroborate the results and added also the analytical treatment of asynchronous dynamical updates. The manuscript improved in quality and I think it is worth to be published in Nature Comm.

A technical point: I think the legend $\Delta J=0.2\dots 1.2$ in Fig 6 should be substituted with $\Delta H=...$